# CiteME: Can Language Models Accurately Cite Scientific Claims?

Ori Press*[1,4], Andreas Hochlehnert*[1,4], Ameya Prabhu[1,4], Vishaal Udandarao[1,3,4], Ofir Press‡[2], and Matthias Bethge‡[1,4]

[1]Tübingen AI Center, University of Tübingen
[2]Princeton Language and Intelligence, Princeton University
[3]University of Cambridge
[4]Open-Ψ (Open-Sci) Collective

## Abstract

Thousands of new scientific papers are published each month. Such information overload complicates researcher efforts to stay current with the state-of-the-art as well as to verify and correctly attribute claims. We pose the following research question: Given a text excerpt referencing a paper, could an LM act as a research assistant to correctly identify the referenced paper? We advance efforts to answer this question by building a benchmark that evaluates the abilities of LMs in citation attribution. Our benchmark, CiteME, consists of text excerpts from recent machine learning papers, each referencing a single other paper. CiteME use reveals a large gap between frontier LMs and human performance, with LMs achieving only 4.2-18.5% accuracy and humans 69.7%. We close this gap by introducing CiteAgent, an autonomous system built on the GPT-4o LM that can also search and read papers, which achieves an accuracy of 35.3% on CiteME. Overall, CiteME serves as a challenging testbed for open-ended claim attribution, driving the research community towards a future where any claim made by an LM can be automatically verified and discarded if found to be incorrect.

## 1 Introduction

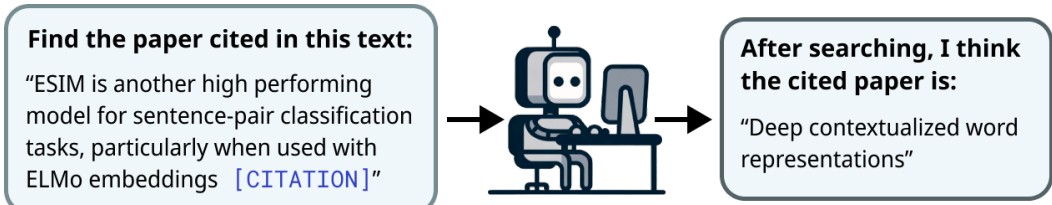

**Find the paper cited in this text:**

"ESIM is another high performing model for sentence-pair classification tasks, particularly when used with ELMo embeddings [CITATION]"

**After searching, I think the cited paper is:**

"Deep contextualized word representations"

Figure 1: **Example of a CiteME instance.** The input (left) is an excerpt from a published paper with an anonymized citation; the target answer (right) is the title of the cited paper.

Scientific discoveries are advancing at an ever-growing rate, with tens of thousands of new papers added just to arXiv every month [4]. This rapid progress has led to information overload within communities, making it nearly impossible for scientists to read all relevant papers. However, it

---

*/‡ shared first/last authorship

Code: github.com/bethgelab/CiteME, Dataset: huggingface.co/datasets/bethgelab/CiteME
Correspondence to {ori.press, andreas.hochlehnert}@bethgelab.org

38th Conference on Neural Information Processing Systems (NeurIPS 2024) Track on Datasets and Benchmarks.

remains a critical scholarship responsibility to check new claims and attribute credit to prior work accurately. Language models (LMs) have shown impressive abilities as assistants across tasks [25], which leads us to explore the following task in this paper: *Can language models act as research assistants to help scientists deal with information overload?*

We make progress towards answering this question by evaluating the abilities of LMs in citation attribution [27, 59]. Given a text excerpt referencing a scientific claim, *citation attribution* is the task in which a system is asked to fetch the title of a referenced paper, as illustrated in Figure 1.

Current benchmarks are collected automatically, which leads to the dominance of ambiguous or unattributable text excerpts that make overly broad claims or are not used as evidence for any specific claim, as shown in Table 1. Furthermore, these benchmarks typically frame citation attribution as a retrieval task from a small set of pre-selected papers where only paper titles and abstracts can be viewed, not the full paper's content important for citation attribution [22, 50].

Table 1: Percentage of reasonable, ambiguous, unattributable, and trivial excerpts across 4 citation datasets, as labeled by human experts. For a detailed breakdown of every analyzed sample, see Appendix A.

|  | Reasonable [%] | Ambiguous [%] | Unattributable [%] | Trivial [%] |
|---|---|---|---|---|
| FullTextPeerRead [42] | 24 | 26 | 34 | 16 |
| ACL-200 [9, 58] | 26 | 42 | 18 | 14 |
| RefSeer [40, 58] | 24 | 28 | 32 | 16 |
| arXiv [33] | 10 | 50 | 30 | 10 |
| Average | 21 | 36.5 | 28.5 | 14 |

To address these issues, we introduce *CiteME* (Citation for Model Evaluation), the first *manually curated* citation attribution benchmark with text excerpts that unambiguously reference a single paper. CiteMe's use of only unambiguous text excerpts eliminates the subjectivity that characterizes other benchmarks.

To evaluate CiteME, we conduct benchmark tests that focus on *open-ended* citation attribution. Human evaluators confirm the lack of ambiguity, achieving 69.7% accuracy while taking just 38.2 seconds on average to find the referenced papers. The current state-of-the-art system, SPECTER2 [77], experiences 0% accuracy on CiteME, highlighting the real-world difficulties of LM-based citation attribution. Similarly, current frontier LMs achieve performance of 4.2-18.5%, substantially beneath human performance. We conclude that current LMs cannot reliably link scientific claims to their sources.

To bridge this gap, we introduce CiteAgent, an autonomous system built on top of the GPT-4o [1] LM and the Semantic Scholar search engine [46]. CiteAgent can search for and read papers repeatedly until it finds the referenced paper, mirroring how scientists perform this scholarship task to find targeted papers. CiteAgent correctly finds the right paper 35.3% of the time when evaluated on CiteME.

In summary, our main contributions are:

- CiteME, a challenging and human-curated benchmark of recent machine learning publications that evaluates the abilities of LMs to correctly attribute scientific claims. CiteME is both natural and challenging, even for SoTA LMs.

- CiteAgent, an LM-based agent that uses the Internet to attribute scientific claims. Our agent uses an existing LM without requiring additional training. It also uses a search engine, which makes it applicable to real-world settings and differentiates it from systems that can search only within a predetermined corpus of papers.

Future work that improves the accuracy of CiteME may lead to systems that can verify *all* claims an LM makes, not just those in the ML research domain. This could reduce the hallucination rate [92] and increase factuality [6] of LM-generated text.

## 2 The CiteME Benchmark

We now present the CiteME benchmark, which we differentiate from other citation prediction benchmarks that are automatically curated, *i.e.,* curated without human supervision or feedback in selecting text excerpts [32, 31, 9, 40, 72, 44, 42, 33]. For comparison, we study the quality of excerpts across four popular citation prediction benchmarks (FullTextPeerRead, [42], ACL-200 [9, 58], RefSeer [40, 58], and arXiv [33]). Specifically, we sample 50 excerpts from each dataset and categorize them using the following criteria:

**(1) Attributable vs Unattributable.** The cited paper should provide *evidence for the statement in the text excerpt*, i.e., be an attribution as opposed to a statement that does not clearly refer to supporting evidence. Excerpts that do not follow this criterion are termed *unattributable,* as in the example:

> *For all of our experiments, we use the hyperparameters from* `[CITATION]`*.*

**(2) Unambiguous vs Ambiguous.** The cited text excerpt should not be overly broad. The ground truth cited papers should clearly be the *only possible reference* for the claim in the text excerpt. Excerpts that do not follow this criterion are termed *ambiguous*, as in the example:

> `[CITATION1, CITATION2]` *explored paper recommendation using deep networks.*

**(3) Non-Trivial vs Trivial.** The text excerpt should not include author names or title acronyms, which simply tests LM memorization and retrieval. Excerpts that do not follow this criterion are termed *trivial*, as in the example:

> *SciBERT* `[CITATION]` *is a BERT-model pretrained on scientific texts.*

**(4) Reasonable vs Unreasonable.** The text excerpt should be attributable, unambiguous and non-trivial. We term excerpts that do not follow this criterion *unreasonable*, but we categorize them according to the underlying issue (e.g., unattributable, ambiguous, or trivial). An example of a reasonable excerpt is:

> *We use the ICLR 2018–2022 database assembled by* `[CITATION]`*, which includes 10,297 papers.*

In Table 1 (left), we demonstrate that most samples from all four datasets lack sufficient information for humans to identify the cited paper and are often labeled as ambiguous or unattributable. Additionally, an average of 17.5% of the samples are tagged as trivial because they include the title of the paper or its authors directly in the excerpt. Excerpts also frequently have formatting errors, making some nearly unreadable (see examples in Appendix A). Past work also notes similar artifacts [33, 42, 58], further supporting our claims. This analysis leads us to contend that performance on existing citation benchmarks might not reflect real-world performance of LM research assistants.

In response to these deficiencies, we created CiteME, a new benchmark with human expert curation for unambiguous citation references. CiteME contains carefully selected text excerpts, each containing a single, clear citation to ensure easy and accurate evaluation.

**Curation.** A team of 4 machine learning graduate students, henceforth referred to as "experts", were responsible for collecting text excerpts. The experts were instructed to find samples that (1) referenced a single paper and (2) provided sufficient context to find the cited paper with scant background knowledge. Each sample was checked for reasonableness; only those deemed reasonable by two or more experts were retained. Some excerpts were slightly modified to make them reasonable.

**Filtering Out the Easy Instances.** To ensure that CiteMe is a challenging and robust dataset, we remove all dataset instances that GPT-4o can correctly answer. Filtering datasets by removing the samples that a strong model can correctly answer was previously done in Bamboogle [71] and the Graduate-Level Google-Proof Q&A Benchmark [73]. In our filtering process, GPT-4o was used with no Internet access or any other external tools. Therefore, it could answer only correctly specified papers that it memorized from its training process. We ran each sample through GPT-4o five times to cover its different outcomes. In the end, we filtered out 124 samples, leaving 130 samples in total.

**Human Evaluation.** To ensure that our benchmark instances are not unsolvable, we evaluate human performance on them. Using a random subset of 100 samples, we asked a group of 20 experts, who were not part of benchmark construction, to perform the task of finding the referenced papers given only the excerpt, with each expert given 5 random samples from CiteME and a maximum of two minutes to solve each instance (similar to [47]). We observe that the experts found the correct citation 69.7% of the time, spending an average of only 38.2 seconds to do so. Note that this accuracy number

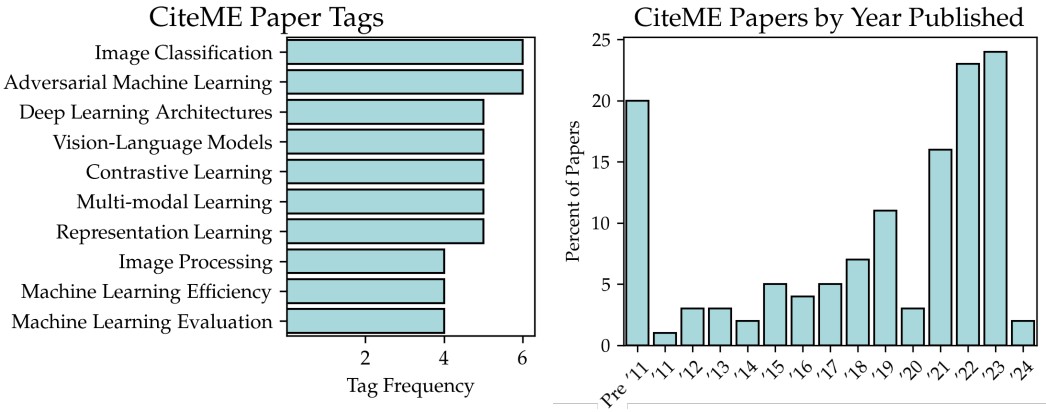

Figure 2: *(Left)* The top 10 most frequent labels of papers in CiteME, as identified by GPT-4. Overly broad tags like "Machine Learning" or "Deep Networks" were excluded (see Appendix D for details). *(Right)* Most excerpts in CiteME are from recent papers.

does not represent the maximum-possible human performance since our annotators were limited to two minutes per question for budget reasons. Human accuracy may rise even higher given more time per instance. To check the experts' consistency, five more experts were asked to solve the same instances previously answered by the original experts. In 71% of the cases, both experts agreed on the answer, and at least one expert got to the right answer in 93% of cases.

**Are 130 questions sufficient to evaluate LMs?** Though traditional machine learning benchmarks usually contain thousands or even millions of test samples, recent work [17, 71, 74, 86] shows that LM benchmarks can include only 100-200 samples and remain insightful. HumanEval [17], for example, which consists of 164 programming problems, is among the most influential LM datasets today, appearing in virtually every SoTA LM paper recently published [66, 1, 81, 19]. Similarly, Bamboogle [71] contains 125 questions, DrawBench [74] contains 200 instances, and Plot2Code [86] contains 132 questions. This is in line with [70, 69], who show that benchmarks with many samples can be reduced to around 100 samples without sacrificing their utility. In addition, smaller benchmarks are advantageous because they are both cheaper to evaluate and impose a less significant environmental impact [76].

## 3   CiteAgent

We now describe CiteAgent, an LM-based system that we built to mimic researcher performance of open-ended citation attribution. A researcher seeking the correct attribution for a claim might use a search engine, read several papers, refine the search query, and repeat until successful. To allow CiteAgent to perform these actions, we built it to use Semantic Scholar to search for and read papers. Unless specified otherwise, we refer to CiteAgent with the GPT-4o backbone simply as CiteAgent throughout this paper.

Given a text excerpt, we prompt CiteAgent to perform one of a fixed set of custom commands and provide the output that the given command generated. CiteAgent then gives its rationale before performing another action, following [90, 88]. Figure 3 shows this process. We now describe the starting prompt and custom agent commands.

**Prompt.** Our prompt includes the task description, descriptions of available commands, and a demonstration *trajectory*, i.e., the series of actions that the system executes while solving an instance [90, 88]. The trajectory includes searching, reading a paper, and searching again (see Figure 4). We model our prompt on the SWE-Agent prompt [88].

**Agent Commands.** CiteAgent can respond to three custom commands (see Table 2). It always begins by executing the `search` command (sorting by relevance or citation count), which searches Semantic Scholar for a query and returns top results in a sorted order. After searching, CiteAgent can either search again, `read` one of the listed papers, or `select` a paper. It can perform up to 15 actions for every sample. Once a `select` action is taken, the session ends, and the selected paper is recorded.

Table 2: Commands available to the model using our system.

| Command | Description |
|---|---|
| search(query, sort) | Searches for a query; sorts results by relevance or by citation count; returns a list of papers, where each item consists of the paper ID, title, number of citations, and abstract. |
| read(ID) | Returns the full text of a paper, including title, author list, abstract, and the paper itself. |
| select(ID) | Selects a paper from the search results as the answer. |

**Search.** CiteAgent initiates a search command by querying Semantic Scholar [46]. We chose the Selenium API [63] over the Semantic Scholar API due to the former's significantly better re-ranked queries and its ability to provide a uniform interface for both our model and human trajectory annotators.

Selenium also lets us access features such as sorting search results by relevance and citation count, which our human trajectory annotators found particularly valuable.

To ensure correctness, we filter out search results published after the excerpt's source paper, and the source paper itself. We then give CiteAgent the top 10 search results, which include paper id, title, abstract, and citation count.

**Read.** Read command execution causes CiteAgent to retrieve the open-access PDF corresponding to the selected paper from Semantic Scholar. Using the PyPDF2 library [29], our system extracts the text from the PDF, excluding visual figures. It then presents the text to CiteAgent, which generates a thought and a new command. If an open-access PDF link is unavailable, CiteAgent returns a message to that effect. We note that due to the limited context length of 8K tokens in the LLaMA-3 LM, we excluded the read action when using that model.

**Select.** Select command execution causes CiteAgent to choose a paper to attribute to the input text excerpt, which ends the run. If the number of actions reaches 14, CiteAgent is prompted to make a selection, forcefully concluding the run. This design choice ensures that all runs complete within a finite time and budget.

## 4 Experiment Setup

Below, we provide detailed implementation information for the baseline models and the various CiteAgent configurations we used for our evaluations.

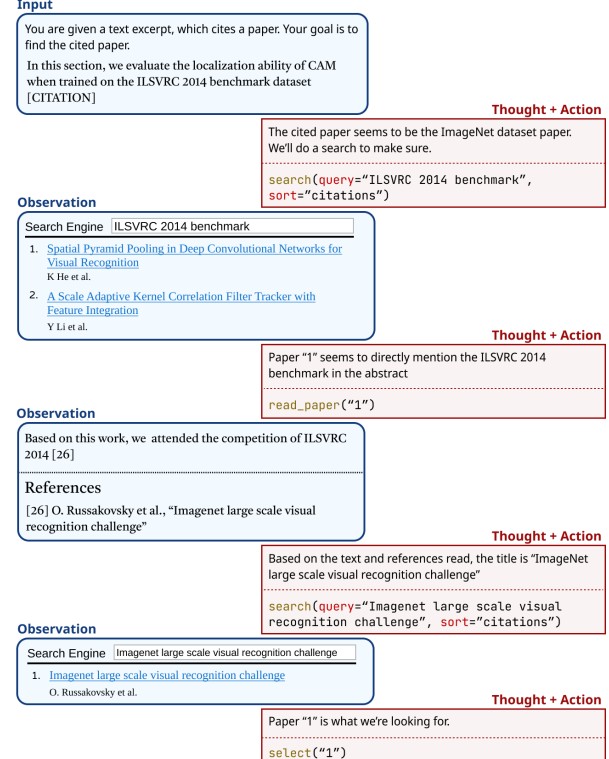

Figure 3: The demonstration trajectory we gave CiteAgent in the prompt.

**SPECTER Models.** We present the results of SPECTER [21] and SPECTER2 [77] on CiteME as our baselines. SPECTER [21] encodes robust document-level representations for scientific texts,

achieving high performance on citation prediction tasks without the need for fine-tuning. We use the Semantic Scholar SPECTER API[1] to embed the input text excerpts and the Semantic Scholar Datasets API[2] to embed all papers on Semantic Scholar, using these embeddings as our retrieval set.

SPECTER2 models [77] introduce task-specific representations, each tailored to different tasks. For our experiments, we use the base customization of SPECTER2 from Hugging Face[3] to embed text excerpts and the Semantic Scholar Datasets API to similarly embed all papers on Semantic Scholar, forming our retrieval set. We apply an exact kNN [53] match to identify the closest embedding, computing the cosine similarity between the embeddings of text excerpt and all available papers (title and abstract). Using exact kNN matches ensures no approximations/errors are introduced while matching queries. We embed the query text excerpt as title only and both title and abstract, but that did not change the performance of the SPECTER models.

**CiteAgent.** We run the CiteAgent system with three SoTA LMs as backbones: GPT-4o [1], Claude 3 Opus [3], and LLaMa-3-70B [81]. We additionally ablate over three classes of commands (Table 2):

1. **Search and Read.** The model can perform both search and read commands.
2. **Search Only.** The model is not allowed to read papers but can perform searches.
3. **No Commands.** The model operates with no access to the interface for actions like searching and reading.

Each class of actions is evaluated with and without demonstrations trajectories in the prompt, resulting in six configurations per LM. With three LMs, two action classes, and the option to include or exclude demonstrations, we present a total of 12 CiteAgent ablations. We exclude LLaMa with both Search and Read because its context length is limited to 8k tokens. For all experiments, we use a temperature of 0.95, following Yang et al. [88], and provide our detailed prompts in Appendix E.

## 5 Results

Table 3: Performance of LMs (using our system) and retrieval methods on CiteME, summarized.

|  | GPT-4o | LLaMA-3-70B | Claude 3 Opus | SPECTER2 | SPECTER1 |
|---|---|---|---|---|---|
| Accuracy [%] | **35.3** | 21.0 | 27.7 | 0 | 0 |

We present the evaluation results of the CiteME benchmark in Table 3. Our best model, CiteAgent (GPT-4o, search and read commands, and a demonstration in the prompt) achieves 35.3% accuracy, while the previous state-of-the-art models, SPECTER2 and SPECTER, achieve 0%. Human performance on the same task is 69.7% accuracy, with less than a minute of search time, indicating that a significant 34.4% gap remains.

Table 4: Accuracy (in %) of LMs and retrieval methods on CiteME. We test how the available commands and prompt demonstrations affect CiteME performance. LLaMA's context window is too small and therefore incompatible with the read command.

| | | | Method | | | | |
|---|---|---|---|---|---|---|---|
| | | | GPT-4o | LLaMA-3-70B | Claude 3 Opus | SPECTER2 | SPECTER |
| **Commands** | **No Commands** | w/o Demo | 0 | 4.2 | 15.1 | 0 | 0 |
| | | w/ Demo | 7.6 | 5.9 | 18.5 | – | – |
| | **Search Only** | w/o Demo | 26.1 | **21.0** | 26.1 | – | – |
| | | w/ Demo | 29.4 | 2.5 | 27.7 | – | – |
| | **Search and Read** | w/o Demo | 22.7 | N/A | **27.7** | – | – |
| | | w/ Demo | **35.3** | N/A | 26.1 | – | – |

**Performance across Language Models.** Comparing the performance of LMs across columns in Table 4, GPT-4o demonstrates the highest accuracy when it has access to both read and search

---

[1] https://github.com/allenai/paper-embedding-public-apis
[2] https://api.semanticscholar.org/api-docs/datasets
[3] https://huggingface.co/allenai/specter2

commands, outperforming other LMs by a wide margin. This finding aligns with previous research [88], which shows that GPT-4 powered agents excel in solving software issues. Notably, GPT-4o achieves high performance across settings even though CiteME consists exclusively of samples that GPT-4o cannot predict correctly without commands; its 0% performance without commands and demonstration trajectory is by design. However, LMs outperforming the SPECTER models purely by autoregressive generation provides evidence that LMs act as implicit knowledge bases with sufficient capacity [68].

**Peformance across Demonstrations.** Comparing the performance between w/o Demo and w/ Demo rows in Table 4, we observe that LLaMA and Claude surprisingly perform worse when provided with a demonstration trajectory in the prompt. This may be due to the increased prompt length, which complicates the detection of important information [52]. LLaMA-3-70b incurs a performance drop to 2.5% due to combined history extending beyond its context length, resulting in errors. However, GPT-4o effectively utilizes demonstrations, which improves its accuracy.

**Performance across Commands.** GPT-4o is the only LM whose accuracy improves with access to more commands, allowing it to read full papers. CiteAgent with GPT-4o creatively uses its commands across test samples, demonstrating command behaviors not shown in the demonstration trajectory (see Figure 4). It frequently refines its searches based on previous results and occasionally reads multiple papers before making a selection. In contrast, Claude 3 Opus is less effective in utilizing additional commands, likely due to difficulties in detecting important information [52].

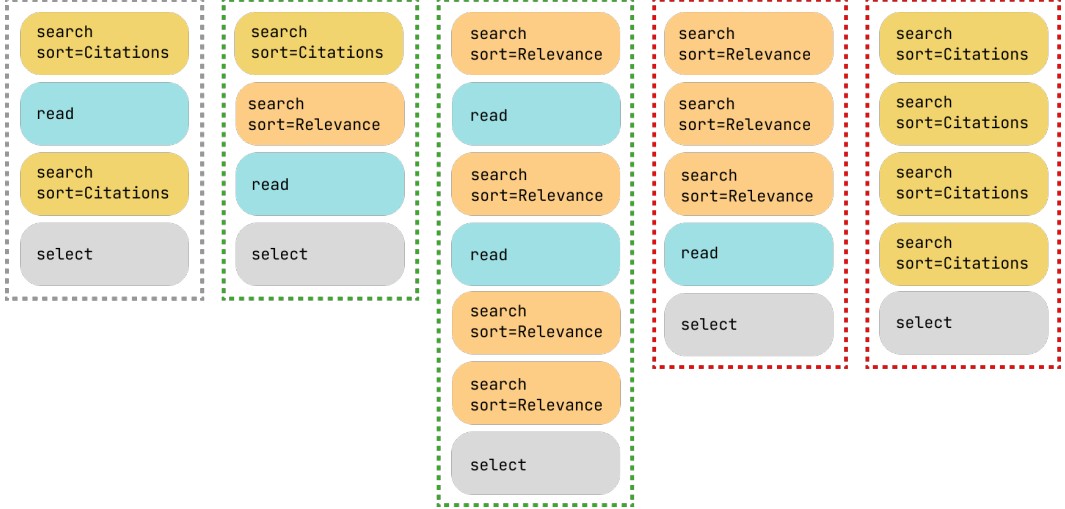

Figure 4: Five CiteAgent trajectories on five different samples. CiteAgent often exhibits behavior not shown in the demonstration given in the prompt, for example: searching by citation count and then by relevance, and searching multiple times in a row. Gray dotted box: prompt demonstration; green dotted boxes: CiteAgent succeeds; red dotted boxes: CiteAgent fails.

## 5.1 Error Analysis

To better identify CiteAgent's shortcomings, we analyze 50 randomly chosen CiteME samples from the best performing CiteAgent (using the GPT-4o backbone, with demonstrations, Search and Read commands) failed to solve correctly. We classify each error into three types based on CiteAgent's searches, its predicted paper and the justification provided:

**Error Type 1: Misunderstands the Excerpt.** This category accounts for 50% of the errors. It occurs when CiteAgent focuses on irrelevant parts of the excerpt or omits critical details. For example, in the following excerpt:

> *The pioneering work of Reed et al. [37] approached text-guided image generation by training a conditional GAN* `[CITATION]`*, conditioned by text embeddings obtained from a pretrained encoder.*

CiteAgent searches for "Reed text-guided image generation conditional GAN" instead of "conditional GAN". It mistakes "Reed" as relevant to the current citation although it pertains to the previous one.

**Error Type 2: Understands the Excerpt but Stops Prematurely.** In 32% of cases, CiteAgent searches for the correct term, but it stops at a roughly matching paper instead of the exact match. For example, in the following excerpt:

> *Using Gaussian noise and blur, [CITATION] demonstrate the superior robustness of human vision to convolutional networks, even after networks are fine-tuned on Gaussian noise or blur.*

CiteAgent found a paper comparing human and machine robustness but missed that it did not cover fine-tuned networks. Notably, this paper referenced the correct target paper, meaning CiteAgent could have found the right answer with just one more step if it had properly understood the paper it was reading. Moreover, in 12.5% of such cases, the correct paper appeared in the search results but was not chosen by CiteAgent.

**Error Type 3: Finds the Correct Citation but Stops Prematurely.** The last 18% of errors occur when CiteAgent reads an abstract or paper and finds the correct citation; however, instead of doing another search, it selects the paper that cites the correct citation and stops searching. For example, in the following excerpt:

> `[CITATION]` *investigates transformers' theoretical expressiveness, showing that transformers cannot robustly model noncounter-free regular languages even when allowing infinite precision.*

CiteAgent finds a paper discussing the target paper and reports it, but it stops at the citing paper instead of searching for the correct target paper. For instance, it reports: *".. specifically mentioning Hahn's work on transformers' classification decisions becoming ineffective over longer input strings. This fits well with the description in the excerpt.."* but it selects the citing paper instead of finding Hahn's work, which is the correct target paper.

**Technical Errors.** Aside from comprehension errors that stem from a lack of understanding an excerpt, 5.8% of runs encountered technical issues. Occasionally, the LM formats responses incorrectly, making them unparseable by the system. Additionally, the Semantic Scholar API has inconsistencies, such as not providing open access PDF links when available or linking to non-existent web pages. Further details on these technical errors are provided in Appendix F.

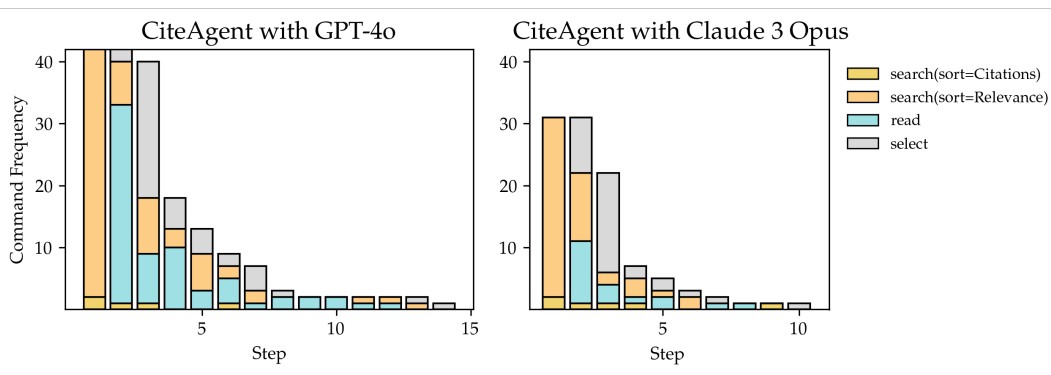

Figure 5: CiteAgent trajectories on samples that were correctly predicted reveals differences in model behavior. GPT-4o reads more frequently than Claude 3 Opus and can correctly predict papers even after performing many actions.

## 5.2 Analyzing the Succesful Runs

Manually examining the instances that were correctly predicted by GPT-4o and Claude 3 Opus (Figure 5) provides insights into how the LMs use commands they were given. First, we confirm the results presented in Table 4: GPT-4o frequently reads papers before it correctly predicts a citation. Second, when both LMs correctly predict a paper, they usually take just 5 steps or fewer to do so. This could stem from LMs loss of important details when given a long context window [52].

CiteAgent's trajectories on CiteME enable us to analyze the shortcomings of GPT-4o and other SoTA LMs. These range from understanding fine details in text (Type 1 and Type 2 Errors), to not completely understanding the task (Type 3 Errors), to being unable to use commands (Technical

Errors). Correcting these errors could improve the utility of LMs on CiteME and for other related tasks.

## 5.3 Benchmarking Reasoning Capability Improvements with Latest Models

Table 5: Accuracy (in %) of newly released LM models on CiteME.

| | | | Method | | |
| | | Claude-3.5-Sonnet | LLaMa-3.1-70B | o1-mini | o1-preview |
|---|---|---|---|---|---|
| **No Commands** | w/o Demo | 8.4 | 3.4 | 16.0 | 38.7 |
| | w/ Demo | 9.2 | 8.4 | 10.9 | – |
| **Search Only** | w/o Demo | 36.1 | **29.4** | 25.2 | – |
| | w/ Demo | **43.7** | **29.4** | 32.8 | – |
| **Search and Read** | w/o Demo | 37.0 | 22.7 | 26.9 | – |
| | w/ Demo | 40.3 | 27.7 | **34.5** | **61.3** |

We compare the latest LLMs on the CiteME benchmark (Table 5) and find that Claude 3.5 Sonnet outperforms the previous best, Claude 3 Opus. This improvement stems from better generalization, as Sonnet achieves 9.2% without internet access, compared to Opus' 18.5%. Similarly, LLaMa-3.1-70B shows significant gains of 8% over LLaMa-3.0-70B, highlighting enhanced reasoning capabilities. However, GPT-o1, while performing well on CiteME, appears to have memorized 38.7% of the dataset, making its 61.3% benchmark performance less clear in terms of true improvement compared to GPT-4o.

# 6 Related Work

Recent work has made substantial progress in developing methods and datasets to assist researchers in paper writing and literature review [8, 12, 87] or act as tutors [18]. Early work [48, 56] showed that researchers automatically retrieved topics and papers considered highly relevant to their work. Other studies included methods that assist researchers in finding new ideas [34], understanding certain topics [62], provide expert answers backed up by evidence [55] or clarifying a paper's related work by supplementing it with more information and focus [15, 67].

Closer to our line of research, prior studies developed methods for substantiating specific claims using evidence from published papers [75, 83, 85, 84, 91, 24, 39, 45]. Retrieval-augmented LMs [49, 11, 30] are also popularly used to ground claims with real-world evidence (see [60] for a survey). Chen et al. [16] built a web-based retrieval-augmented pipeline for fact verification; this contrasts with methods that use a static dataset for claim retrieval and verification [36, 5]. Concurrent to this work, Ajith et al. [2] build a retrieval benchmark consisting of questions about discoveries shown in specific machine learning papers.

Paper discovery is a crucial component of systems that automate scientific research as shown in [10, 47, 54, 61, 78]. CiteME plays an important role in developing better tools for paper discovery, and provides a way to effectively measure their efficiency. Currently, these systems are tested as a whole, without isolating the tools responsible for scientific discovery. CiteME allows us to evaluate components within them independently – and we discover that current LM Agents are not yet ready for automated paper discovery, leading to serious gaps in end-to-end automated research pipelines.

In addition, most existing LM benchmarks are saturated, with most LMs scoring 80-95% on them [43, 38, 20]. There is a need in the AI community to show what properties LMs currently lack, to show LM developers what aspects they should work on. On CiteME, the best LMs get less than 40%, clearly indicating to developers an important task that they could improve LMs on, while also providing an indicator they can use to track progress.

**Context-aware Recommendation.** Relevant to our research focus, [57, 64, 37] take as input documents or parts thereof and recommend papers that are likely to be cited, often referred to as *context-aware citation recommendation* [51, 26, 89, 28, 42, 65, 33]. The text inputs we use in CiteME resemble those used in [42, 65, 80], which contain a few sentences with a masked out citation. However, CiteME differs because it uses excerpts containing only one unambiguous citation, making

the context sufficient to identify the cited paper. Furthermore, our work explores agents with access to real-time paper information through tools like Semantic Scholar. This is crucial for real-time use since thousands of new papers are indexed by arXiv monthly (e.g., 8,895 papers in March 2024 under the `cs` category) [4]. Most previous approaches would be impractical due to the need for retraining with every new paper issuance.

**Citation Attribution Datasets.** A variety of datasets contain text excerpts from scientific papers and corresponding citations [32, 31, 9, 40, 72, 44, 42, 33]. There are many crucial distinctions between the aforementioned datasets and CiteME, with the main one being that CiteME is composed of manually selected excerpts that clearly reference a paper. To our best knowledge, *CiteME is the only dataset that reports human accuracy on the benchmark.*

Additionally, the excerpts in CiteME are mostly taken from papers published in the last few years (see Figure 2), whereas other datasets contain older papers. For example, the arXiv dataset [33] includes papers from 1991-2020, and FullTextPaperRead [42] contains papers from 2007-2017. This currency is particularly relevant in rapidly evolving fields like machine learning. The key distinction between the dataset and methods we present compared to previous works is their *real life applicability*. Our agent is based on SoTA LMs, needs no extra training, and can use a search engine, all of which make it easily applicable to real-world settings.

# 7   Conclusion

This work introduces a citation attribution benchmark containing manually curated text excerpts that unambiguously refer to a single other paper. We posit that methods that succeed on CiteME are likely to be highly useful in assisting researchers with real-world ML-specific attribution tasks but also generally useful in finding sources for generic claims. Further, our CiteAgent autonomous system can search the Internet for and read papers, which we show to significantly enhance the abilities of LMs on CiteME. We anticipate that this work will lead to LMs that are more accurate research assistants in the vital scholarship tasks of attribution.

# Author Contributions

The project was initiated by Andreas Hochlehnert and Ori Press, with feedback from Ameya Prabhu, Ofir Press, and Matthias Bethge. The dataset was created by Ori Press and Ameya Prabhu, with help from Vishaal Udandarao and Ofir Press. Experiments were carried out by Andreas Hochlehnert, with help from Ameya Prabhu. All authors contributed to the final manuscript.

# Acknowledgements

The authors thank the International Max Planck Research School for Intelligent Systems (IMPRS-IS) for supporting Ori Press, Andreas Hochlehnert, and Vishaal Udandarao. Andreas Hochlehnert is supported by the Carl Zeiss Foundation through the project "Certification and Foundations of Safe ML Systems". Matthias Bethge acknowledges financial support via the Open Philanthropy Foundation funded by the Good Ventures Foundation. Vishaal Udandarao was supported by a Google PhD Fellowship in Machine Intelligence. Matthias Bethge is a member of the Machine Learning Cluster of Excellence, funded by the Deutsche Forschungsgemeinschaft (DFG, German Research Foundation) under Germany's Excellence Strategy – EXC number 2064/1 – Project number 390727645 and acknowledges support by the German Research Foundation (DFG): SFB 1233, Robust Vision: Inference Principles and Neural Mechanisms, TP 4, Project No: 276693517. This work was supported by the Tübingen AI Center. The authors declare no conflicts of interests.

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

# A Excerpts from Citation Datasets

To demonstrate the problematic nature of automatically sourced text excerpts, we randomly choose 10 excerpts from FullTextPeerRead, ACL-200, RefSeer, and arXiv. We tag each sample chosen with one of 4 tags, as summarised in Table 1 in the main paper. We show each sample as it appears verbatim, using the datasets that appear in the official repository[4] of Gu et al. [33].

**ACL-200**  [9, 58]

- m which the data was extracted (original). We used a combination of automatic (e.g. BLEU–4 (OTHERCIT), METEOR (OTHERCIT)) and human metrics (using crowdsourcing) to evaluate the output (see generally, TARGETCIT . However, in the interest of space, we will restrict the discussion to a human judgment task on output preferences. We found this evaluation task to be most informative for system improvement. The ta
  **Unattributable**

- n Section 2 that it is more difficult to extract keyphrases correctly from longer documents. Second, recent unsupervised approaches have rivaled their supervised counterparts in performance (OTHERCIT; TARGETCIT b). For example, KP-Miner (OTHERCIT), an unsupervised system, ranked third in the SemEval-2010 shared task with an F-score of 25.2, which is comparable to the best supervised system scoring 27.5. 5 An
  **Ambiguous**: The citation is ambiguous by definition, as the excerpt cites more than one paper.

- rams include unigrams for all feature definitions and bigrams for selected ones. Figure 3b shows a sample of the actual extended set. We use two datasets, one prepared for the CoNLL 2000 shared task ( TARGETCIT and another prepared for the BioNLP/NLPBA 2004 shared task (OTHERCIT). They represent two different tagging tasks, chunking and named entity recognition, respectively. The CoNLL 2000 chunking dataset
  **Trivial**

- ipts were from meetings, seminars and interviews. Some authors have also referred to this phenomenon as Ellipsis because of the elliptical form of the NSU [OTHERCIT, Fern´andez et al., 2004, OTHERCIT, TARGETCIT , OTHERCIT]. While the statistical approaches 336 have been investigated for the purpose of ellipsis detection [Fern´andez et al., 2004, OTHERCIT], it has been a common practice to use rules – syntact
  **Ambiguous**: The citation is ambiguous by definition, as the excerpt cites more than one paper.

- e source language is morphologically poor, such as English, and the target language is morphologically rich, such as Russian, i.e., language pairs with a high degree of surface realization ambiguity ( TARGETCIT . To address this problem we propose a general approach based on bilingual neural networks (BNN) exploiting source-side contextual information. This paper makes a number of contributions: Unlike previ
  **Reasonable**

- n our approach and the one described in (OTHERCIT). Such a similarity is calculated by using the WordNet::Similarity tool (OTHERCIT), and, concretely, the Wu-Palmer measure, as defined in Equation 1 ( TARGETCIT . 2N3 Sim(C1, C2) ? (1) N1 + N2 + 2N3 where C1 and C2 are the synsets whose similarity we want to calculate, C3 is their least common superconcept, N1 is the number of nodes on the path from C1 to C3,
  **Reasonable**

- ch detected image object a visual attribute and a spatial relationship to the other objects in the image. The spatial relationships are translated into selected prepositions in the resulting captions. TARGETCIT used manually segmented and labeled images and introduced visual dependency representations (VDRs) that describe spatial relationships between the image objects. The captions are generated using templ
  **Reasonable**

- ous open source machine translation systems. The widely used Moses system (OTHERCIT) implements the standard phrase-based translation model. Parsingbased translation models

---

[4]https://github.com/nianlonggu/Local-Citation-Recommendation

are implemented by Joshua ( TARGETCIT , SAMT (OTHERCIT), and cdec (OTHERCIT). Cunei (OTHERCIT) implements statistical example-based translation. OTHERCIT and OTHERCIT respectively provide additional open-source implementations of phrase-b
**Trivial**

- and test set, we had about 1000 sentences each with 10 reference translations taken from the NIST 2002 MT evaluation. All Chinese data was re-segmented with the CRF-based Stanford Chinese segmenter ( TARGETCIT that is trained on the segmentation of the Chinese Treebank for consistency. The parser used in Section 3 was used to parse the training data so that null elements could be recovered from the trees.
**Trivial**

- rdering between nodes), their means of creation, and the scoring method used to extract the best consensus output from the lattice (OTHERCIT). In speech processing, phoneme or word lattices (OTHERCIT; TARGETCIT are used as an interface between speech recognition and understanding. Lat1318 Proceedings of the 48th Annual Meeting of the Association for Computational Linguistics, pages 1318–1327, Uppsala, Sweden
**Ambiguous**: The citation is ambiguous by definition, as the excerpt cites more than one paper.

**RefSeer**   [40, 58]

- . Their experiments suggested that view independence does indeed affect the performance of co-training; but that CT, when compared to other algorithms that use labeled and unlabeled data, such as EM ( TARGETCIT ; OTHERCIT), may still prove e#ective even when an explicit feature split is unknown, provided that there is enough implicit redundancy in the data. In contrast to previous investigations of
**Ambiguous**: The citation is ambiguous by definition, as the excerpt cites more than one paper.

- eeded is NP-hard. On the other hand, if the permutation $\pi$ avoids the pattern 1-2-3, no shuffles are needed if $k \geq 5$ (this is the result that every triangle free circle graph is 5-colorable, see again TARGETCIT ). It becomes clear once more why circle graphs "frustrated mathematicians for some years" OTHERCIT, and still continue to do so. 5 Stacking Constraints We finally consider the generalization in which ite
**Reasonable**

- a small number of details they have many things in common, especially the process of motion compensation and the DCT. Due to similar motion compensation the motion vector (MV) can be reused very well TARGETCIT . Furthermore, the equivalent usage of the DCT of block size ? ? makes a transcoder implementation within the DCT-domain possible OTHERCIT. With the standardization of H.264 the task of heterogeneous trans
**Reasonable**

- tioned Transactions ? Lingxiang Xiang Michael L. Scott Department of Computer Science, University of Rochester lxiang, scott@cs.rochester.edu 1. Introduction Twenty years after the initial proposal TARGETCIT , hardware transactional memory is becoming commonplace. All commercial versions to date—and all that are likely to emerge in the near future—are best effort implementations: a transaction may abort a
**Reasonable**

- local values generating a cluster are uniformly distributed in the range of $[\mu_{ij} - \sigma_{ij} \times 0.01, \mu_{ij} + \sigma_{ij} \times 0.01]$. ? Irrelevant feature f ? $j \in S_i$ : We uniformly generate values in the entire range TARGETCIT . We then synthetically generate co-occurrence scores. While the co-occurrence score can be arbitrarily generated, it is non-trivial to decide the ground-truth clusters when featurebased and co-occurr
**Unattributable**

- for visualizing the messagesow between objects in terms of method invocations. The scenario diagrams are generated from event traces and linked to other sources of information. Jerding and colleagues TARGETCIT , OTHERCIT focus on the interactions between program components at runtime. They observed that recurring interaction pattern can be used in the abstraction process for program understanding. The authors d
**Trivial**: Though the cited excerpt cites more than one paper, that author name is given.

- Many multimedia services, such as audio-video conferencing or video playback, have associated with them performance requirements that must be met to guarantee acceptable service to the users. TARGETCIT describes the requirements that some typical applications place on networks. The Tenet Real-Time Protocol Suite [Ferrari92 ] is one approach to providing these real-time performance guarantees in pac
**Unattributable**

- y of the controlled system is jeopardized. Several scheduling paradigms have been developed to support the analysis of a task set and determine if a schedule is feasible, e.g., rate-monotone analysis TARGETCIT . These scheduling paradigms rely on the assumption that the worst-case execution time (WCET) of hard real-time tasks be known a-priori. If the WCET of all tasks is known, it can be determined if a sc
**Reasonable**

- Recommended for acceptance by L. Quan. For information on obtaining reprints of this article, please send e-mail to: tpami@computer.org, and reference IEEECS Log Number TPAMI-0308-1003. æ recovered TARGETCIT , OTHERCIT. Note that these calibration techniques can be used for both central and noncentral catadioptric cameras. 2. Self-calibration. This kind of calibration techniques uses only point correspo
**Ambiguous**: The citation is ambiguous by definition, as the excerpt cites more than one paper.

- ic controller in which a single action is associated with each node, and an observation results in a deterministic transition to a successor node (OTHERCIT; Hansen 1998; TARGETCIT a). In other cases, it is a stochastic controller in which actions are selected based on a probability distribution associated with each node, and an observation results in a probabilistic transition
**Ambiguous**: The citation is ambiguous by definition, as the excerpt cites more than one paper.

**arXiv**  [33]

- In this study we parallelized the computation of gradients to improve the efficiency, and for large datasets further improvements can be obtained by using random minibatches to perform the inversion TARGETCIT . Such a strategy can be applied to any variational inference method (e.g. also ADVI) since variational methods solve an optimization rather than a stochastic sampling problem. In comparison, this st
**Unattributable**

- e been shown to provide superior generative quality, but VAEs have a number of advantages which include outlier robustness, improved training stability and interpretable, disentangled representations TARGETCIT . Disentangled representations are generally conceived to be representations in which each element relates to an independent (and usually semantically meaningful) generative factor OTHERCIT OTHERCIT . Achieving a di
**Reasonable**

- tion (NTF) OTHERCIT . For example, NMF/NTF-based ML methods have been successfully used for analysis of Monte Carlo simulated fission chamber's output signals OTHERCIT , for compression of scientific simulation data TARGETCIT , and for a variety of other applications OTHERCIT . To avoid confusion, we should emphasize that in this paper the term tensor is used to define two different types of mathematical objects. We use tensors t
**Unattributable**

- insight about the generalization to the multipartite scenario, but also since the recovery problem for a tripartite probability distribution given all the three possible bipartite marginals is open OTHERCIT TARGETCIT OTHERCIT . Moreover, moving to the quantum scenario, also the compatibility problem for just a couple of overlapping marginals is open OTHERCIT OTHERCIT . We are then going to assume the set of the two given marginal densit
**Ambiguous**: The citation is ambiguous by definition, as the excerpt cites more than one paper.

- seen that the proxy-SU(3) symmetry suggests N = 116 as the point of the prolate-to-oblate shape/phase transition, in agreement with existing exprerimental evidence OTHERCIT OTHERCIT OTHERCIT OTHERCIT OTHERCIT and microscopic calculations OTHERCIT OTHERCIT TARGETCIT OTHERCIT . Table 1 . Comparison between SU(3) irreps for

U(6), U(10), U(15), and U(21), obtained by the code UNTOU3 OTHERCIT , contained in the relevant U(n) irrep for M valence protons or M valence neutrons. Above
**Ambiguous**: The citation is ambiguous by definition, as the excerpt cites more than one paper.

- h cannot be explained by the traditional expected utility theory. In the context of decision-theoretic systems, Nadendla et al. have presented detection rules employed by prospect theoretic agents in TARGETCIT under different scenarios based on decision costs. In particular, the authors have focused on two types of prospect theoretic agents, namely optimists and pessimists, and have shown that the prospect
**Trivial**: The name of the author of the referenced paper appears in the excerpt.

- .) (3) $\psi(\wedge S)$ does depend on the isotopy class of the collection. Its image in the space $A(\star k 1 ,... ,k\mu )$ , however, does not. These issues, and the above proof, are discussed in full detail in TARGETCIT . We remark that, in the form presented, this theorem does not depend on the two pieces of heavy machinery employed by OTHERCIT -it depends on neither the adapted Kirby-Fenn-Rourke theorem nor the OTHERCIT calculati
**Unattributable**

- ed to follows an addition rule 2ND 2 = analogous to that found for frequency conversion. A series of recent experiments demonstrated a more complex transfer of OAM in the generation of Raman sideband TARGETCIT OTHERCIT OTHERCIT . This process was found to follow a now wellestablished OAM-algebra for Stokes and anti-Stokes orders and was definitively verified through phase measurements in a simultaneous Young double slit e
**Ambiguous**: The citation is ambiguous by definition, as the excerpt cites more than one paper.

- BMD. An important tool to assess the performance of decoding metrics is the generalized mutual information (GMI) OTHERCIT Sec. 2.4 ]. An interpretation of uniform BMD and bit-shaped BMD as a GMI are given in TARGETCIT and OTHERCIT , respectively. In OTHERCIT Sec. 4.2.4 ], the GMI is evaluated for a bit-metric. It is observed that the GMI increases when the bits are dependent. We call this approach shaped GMI. Besides the GMI, oth
**Ambiguous**: The citation is ambiguous by definition, as the excerpt cites more than one paper.

- cay products dilute faster than matter, the expansion rate can be reduced around z $\sim$ 2.3. However, the simplest such model, a dark matter component decaying into dark radiation with constant lifetime TARGETCIT OTHERCIT , is in conflict with observations of the late integrated SachsWolfe effect and lensing power spectrum OTHERCIT OTHERCIT . Moreover, we find $\Omega$ ExDE becomes positive again at z < 1.5. Thus any decaying component mus
**Ambiguous**: The citation is ambiguous by definition, as the excerpt cites more than one paper.

## FullTextPeerRead [42]

- tion function: r=g.The typical training criterion for autoencoders is minimizing the reconstruction error, $\Sigma x \in XL$ with respect to some loss L, typically either squared error or the binary cross-entropy TARGETCIT .Denoising autoencoders are an extension of autoencoders trained to reconstruct a clean version of an input from its corrupted version . The denoising task requires the network to learn representatio
**Ambiguous**: Although [7] is cited, it could be argued that the original paper that used cross entropy as a loss [23] should be used.

- al matrices of parameters, and show that it outperforms the random counterpart when applied to the problem of replacing one of the fully connected layers of a convolutional neural network for ImageNet TARGETCIT . Interestingly, while the random variant is competitive in simple applications , the adaptive variant has a considerable advantage in more demanding applications .The adaptive SELLs, including Adapti
**Trivial**

- eneous information networks. Recently, u peek_meaning:NTF . peek_catcode:NTF a . . published a question answering algorithm that converts a given question into a vector space

model to find the answer TARGETCIT , but, like neural network based models 2013 , the learned model is generally uninterpretable. peek_meaning:NTF . peek_catcode:NTF a . . proposed T-verifier, a search engine based fact checker 2011
**Ambiguous**: The cited paper is [35], while [41] also fits the description given.

- he graph's main component correctly. The state-of-the-art described in gives a lowest value at 58, with the best algorithms around 60, while algorithms regularized spectral methods such as the one in TARGETCIT obtain about 80 errors.The current result should also extend directly to a slowly growing number of communities . It would be interesting to extend the current approach to smaller sized communities or
  **Unattributable**

- amming approach that was used in all other structural tractability results that were known before, and as we have seen this is no coincidence. Instead, B-acyclic #SAT lies outside the STV-framework of TARGETCIT that explains all old results in a uniform way.We close this paper with several open problems that we feel should be explored in the future. First, our algorithm for #SAT is specifically designed for
  **Unattributable**

- our method on a fully-connected network , we compare our method with on this dataset. CIFAR and SVHN dataset, we evaluate our method on three popular network architectures: VGGNet, Net and DenseNet TARGETCIT . The VGGNet is originally designed for ImageNet classification. For our experiment a variation of the original VGGNet for CIFAR dataset is taken from . For Net, a 164-layer pre-activation Net with bo
  **Trivial**

- ars, various probabilistic extensions of description logics have been investigated, see, for instance,.The one that is closest to our approach is the type 1 extension of ALC proposed in the appendixof TARGETCIT . Briefly, This difference is the main reason why the ExpTime algorithm proposed by tz and Schrödercannot be transferred to our setting. It does not suffice to consider the satisfiable types independ
  **Unattributable**

- h we compute through current input and the previous hidden state. The final output of hidden state would be calculated based on memory cell and forget gate.In our experiment we used model discussed in TARGETCIT .t x is feature vector for tth word in a sentence and hl is previous hidden state then computation of hidden and output layer of LSTM would be.Where $\sigma$ is sigmoid activation function, $\star$ is a element
  **Unattributable**

- e use of conditional LSTMs in the generation component of neural network -based dialogue systems which depend on multiple conditioning sources and optimising multiple metrics.ral conversational agents TARGETCIT are direct extensions of the sequence-to-sequence model in which a conversation is cast as a source to target transduction problem.wever, these models are still far from real world applications becau
  **Ambigiuous**: The cited paper is [82], though [79] also fits the description given.

- onsistent with previous findings.As a comparison we also include test performances of a BNN with a Gaussian approximation , a BNN with HMC, and a sparse Gaussian process model with 50 inducing points TARGETCIT . In test-LL metric our best dropout model out-performs the Gaussian approximation method on almost all datasets, and for some datasets is on par with HMC which is the current gold standard for yesian
  **Ambiguous**: The cited paper is [13], while [14] also fits the description given.

### A.1 Automatic Ambiguity Analysis

In addition to the manual analysis above, we conducted an automated analysis of the ambiguous category. Specifically, we identified excerpts that cited multiple papers simultaneously (e.g., `\cite{paper1, paper2, paper3}`) where one of the cited papers is the target. This analysis allowed us to establish a lower bound on ambiguous excerpts across all benchmarks (Table 6). These excerpts can not serve well as questions since they have multiple different correct answers, whereas the respective benchmarks only include one correct target answer (as in CiteME).

Table 6: Dataset ambiguity percentages from an automatic analysis. We note that this is just a lower bound estimate, as the automatic parsing is only able to detect a subset of the ambiguous excerpts. Still, these findings are consistent with our previous results, and show that previous benchmarks contain vast quantities of ambiguous excerpts.

| Dataset | Ambiguous [%] |
|---------|---------------|
| arXiv   | 54.96         |
| ACL     | 27.20         |
| RefSeer | 12.61         |

FulllTextPeerRead automatically deletes all other citations, so this was not possible to do in their case. We have updated Table 1 in the revised draft with the results with the expanded 50-sample sets and included the automatic evaluation data.

# B Additional Comparison to Existing Benchmarks

We additionally compare CiteME to previous benchmarks based on information found in [33]. Importatnly, CiteME differs from previous work in that the query set, from which the answers come from, is by far the largest with 218 million papers. Additionally, CiteME makes the entire paper available to the model, and not just a snippet. These two factors make CiteME able to mimic the experience a research would have when looking for papers.

Table 7: Comparison of previous benchmarks and CiteME based on query set size, availability of full paper text, and date range.

| Dataset | Query Set Size | Full Paper Text | Date Range |
|---|---|---|---|
| FullTextPeerRead [42] | 5K | ✗ | '07-'17 |
| ACL-200 [9, 58] | 20K | ✗ | '09-'15 |
| RefSeer [40, 58] | 625K | ✗ | Unk - '14 |
| arXiv [33] | 1.7M | ✗ | '91-'20 |
| CiteME (Ours) | 218M | ✓ | '08-'24 |

# C CiteAgent Results By Year

Language models may perform better on papers they encountered during training, with a drop in performance on newer papers, leading to better performance from more recently released models. To test this, we compare the results of our CiteAgent on excerpts from papers published before 2024 versus on excerpts from papers published in 2024. We note that the cutoff dates for Claude 3 Opus, Claude 3.5 Sonnet and GPT-4o are August 2023, August 2023 and October 2023 respectively. The results, shown in Table 8, show that this is indeed true for the LMs analyzed in this paper.

Table 8: Accuracy of CiteAgent models (in %) on questions where the target papers were published either before 2024 and during 2024

| Model | Before 2024 | 2024 |
|---|---|---|
| CiteAgent + GPT-4 | 36.99% | 32.61% |
| CiteAgent + Claude 3 Opus | 28.77% | 21.74% |
| CiteAgent + Claude 3.5 Sonnet | 42.47% | 36.96% |

# D Verifying GPT-4 Paper Tags

We asked GPT-4 to generate 3 general tags that describe every paper in CiteME. We manually verify that the tags automatically generated by GPT-4 are overwhelmingly correct. Here, we give a few examples of papers, and their matching tags:

- **Paper Name:** PaLI: A Jointly-Scaled Multilingual Language-Image Model
  **Tags:** Multimodal AI Models, Vision-Language Integration, Scalable Machine Learning

- **Paper Name:** Grokking: Generalization Beyond Overfitting on Small Algorithmic Datasets
  **Tags:** Neural Network Generalization, Deep Learning Optimization, Algorithmic Data Analysis

- **Paper Name:** Minimally distorted Adversarial Examples with a Fast Adaptive Boundary Attack
  **Tags:** Adversarial Machine Learning, Neural Network Security, Robustness Evaluation]

- **Paper Name:** Mamba-R: Vision Mamba ALSO Needs Registers
  **Tags:** Computer Vision Models, Image Processing Techniques, Neural Network Architectures

- **Mass-Editing Memory in a Transformer**
  **Tags:** Language Model Updating, Natural Language Processing, Machine Learning Techniques

## E    Example Trajectory

Please note that the abstracts retrieved in the paper search have been redacted, and only the first two of the ten results are displayed to condense the text. Additionally, the content retrieved by the paper reading action has been abbreviated. The system prompt is omitted for brevity. Complete and exact prompts are available in the published code.

**User:**

```
You are now given an excerpt. Find me the paper cited in the excerpt, using
the tools described above. Please make sure that the paper you select really
corresponds to the excerpt: there will be details mentioned in the excerpt
that should appear in the paper. If you read an abstract and it seems like
it could be the paper we're looking for, read the paper to make sure.
Also: sometimes you'll read a paper that cites the paper we're looking for.
In such cases, please go to the references in order to find the full name
of the paper we're looking for, and search for it, and then select it.

Our implementation for ImageNet follows the practice in [21, 41]. The
image is resized with its shorter side randomly sampled in [256, 480]
for scale augmentation [41]. A 224×224 crop is randomly sampled from
an image or its horizontal flip, with the per-pixel mean subtracted [21].
The standard color augmentation in [21] is used. We adopt batch
normalization (BN) [CITATION] right after each convolution and before
activation.
```

**Assistant:**

```
{
    "reason": "The excerpt mentions batch normalization (BN) and places
               it right after each convolution and before activation.
               I'll search for 'batch normalization' to find the
               appropriate citation.",
    "action": {
        "name": "search_relevance",
        "query": "batch normalization"
    }
}
```

**User:**

```
- Paper ID: 995c5f5e62614fcb4d2796ad2faab969da51713e
   Title: Batch Normalization: Accelerating Deep Network Training
          by Reducing Internal Covariate Shift
   Abstract: <redacted>
   Citation Count: 39617

- Paper ID: 713bbd21bd4780fc92a69645a8bee2a005cc90dd
   Title: Comparison of normalization algorithms for cross-batch
          color segmentation of histopathological images
   Abstract: <redacted>
   Citation Count: 18

<Remaining 8 Results>
```

**Assistant:**

```
{
    "reason": "The paper with ID 995c5f5e62614fcb4d2796ad2faab969da51713e
               titled 'Batch Normalization: Accelerating Deep Network
               Training by Reducing Internal Covariate Shift' is explicitly
```

```
                 about batch normalization and aligns perfectly with the
                 mention of batch normalization in the given excerpt.",
    "action": {
        "name": "read",
        "paper_id": "995c5f5e62614fcb4d2796ad2faab969da51713e"
    }
}
```

**User:**

arXiv:1502.03167v3 [cs.LG] 2 Mar 2015BatchNormalization:
AcceleratingDeepNetworkTrainingb y ReducingInternalCovariateShift
Sergey Ioffe GoogleInc., sioffe@google.comChristianSzegedy GoogleInc.,
szegedy@google.com Abstract TrainingDeepNeuralNetworksiscomplicatedbythefact
that the distributionofeach layer'sinputschangesduring training, as the
parametersof the previouslayers change. This slows down the training by
requiringlower learning

<Remaining Content of the Paper>

**Assistant:**

```
{
    "reason": "The paper Batch Normalization: Accelerating Deep Network
               Training by Reducing Internal Covariate Shift by Sergey
               Ioffe and Christian Szegedy precisely matches the
               description in the excerpt given. It details the
               application of batch normalization, placing it right after
               each convolution and before activation.",
    "action": {
        "name": "select",
        "paper_id": "995c5f5e62614fcb4d2796ad2faab969da51713e"
    }
}
```

# F Technical Errors

In this section, we break down the types of technical errors that can occur when running CiteAgent with a GPT-4o or Claude 3 Opus backbone.

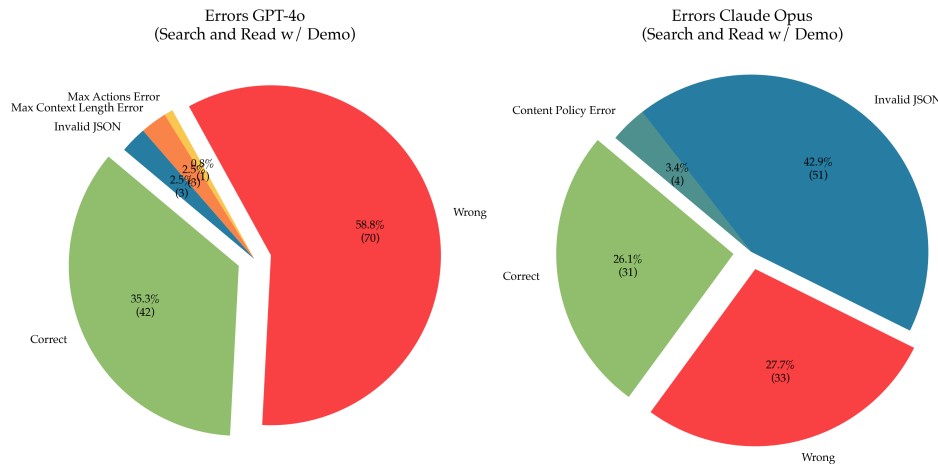

Figure 6: Different technical errors for the CiteAgent with `Search` and `Read` command with Demo comparing the GPT-4o and Claude Opus backbone. Claude Opus has a significantly higher error rate. It struggles to adhere to the expected JSON fromat and in four cases the content filter was triggered.

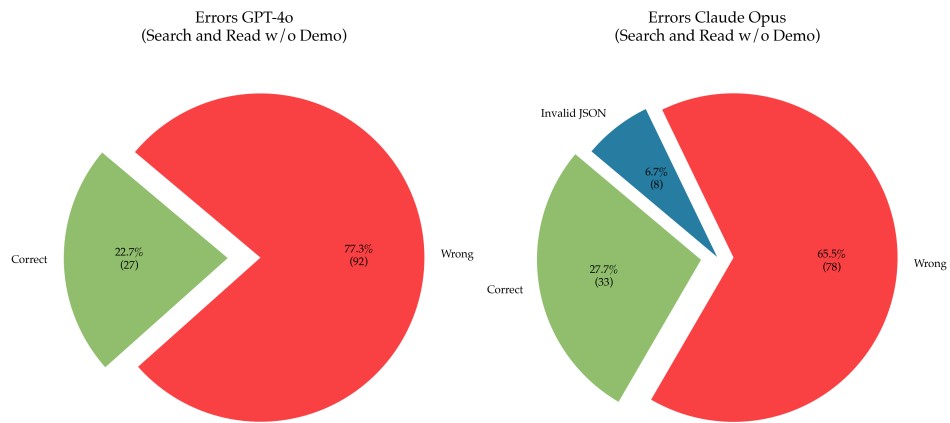

Figure 7: Different technical errors for the CiteAgent with `Search` and `Read` command without Demo comparing the GPT-4o and Claude Opus backbone. Because there is no demo the system prompt is much shorter just containing the task description and the format instructions. One can see that the JSON error rate for Claude Opus is now drastically reduced. GPT-4o also exhibits a smaller error rate but its performance is degraded.

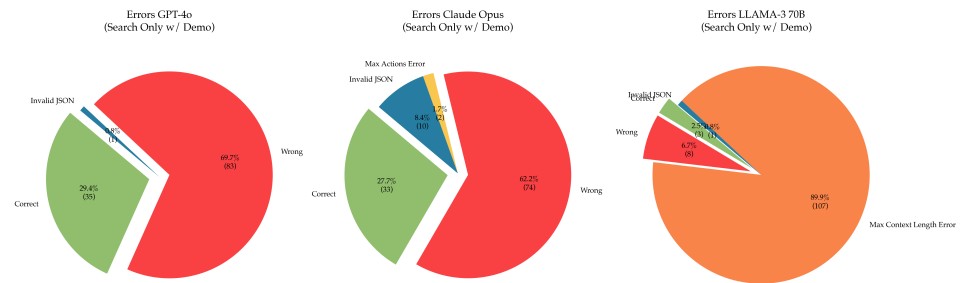

Figure 8: Different technical errors for the CiteAgent with `Search Only` command with Demo comparing the GPT-4o, Claude Opus and LLaMA-3 70B backbone. The system prompt containing the Demo takes up a considerable amount of LLaMA-3's context length therefore just a few actions lead to the model running out of context.

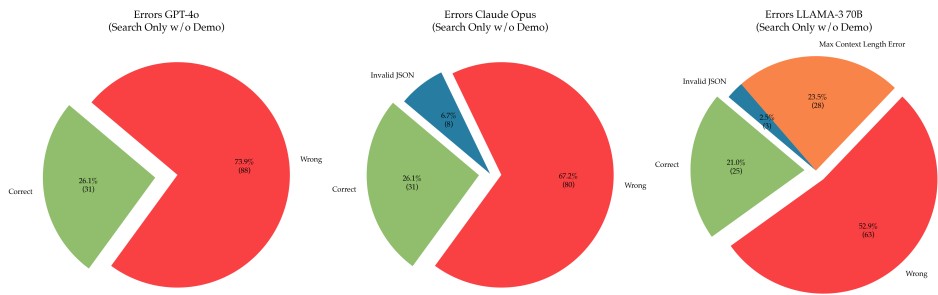

Figure 9: Different technical errors for the CiteAgent with `Search Only` command without Demo comparing the GPT-4o, Claude Opus and LLaMA-3 70B backbone.

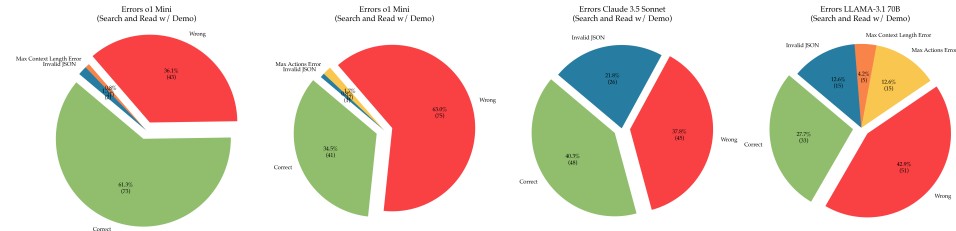

Figure 10: Different technical errors for the CiteAgent with `Search` and `Read` command with Demo comparing the o1-Preview, o1-Mini, Claude 3.5 Sonnet and LLaMA-3.1 70B backbone.

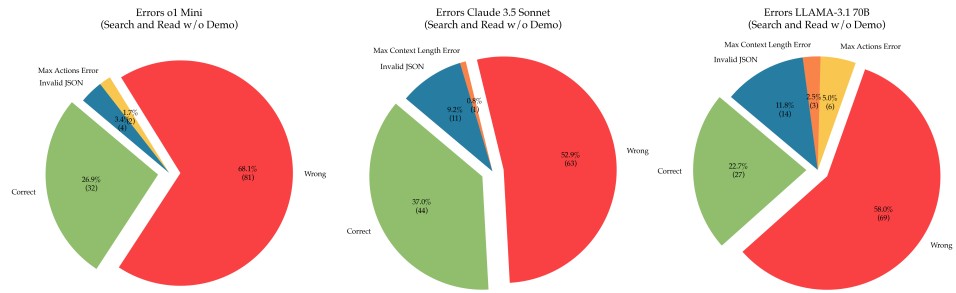

Figure 11: Different technical errors for the CiteAgent with `Search` and `Read` command without Demo comparing the o1-Mini, Claude 3.5 Sonnet and LLaMA-3.1 70B backbone.

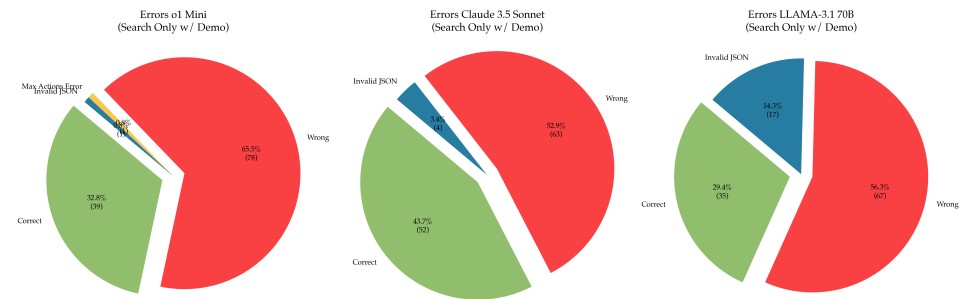

Figure 12: Different technical errors for the CiteAgent with `Search Only` command with Demo comparing the o1-Mini, Claude 3.5 Sonnet and LLaMA-3.1 70B backbone.

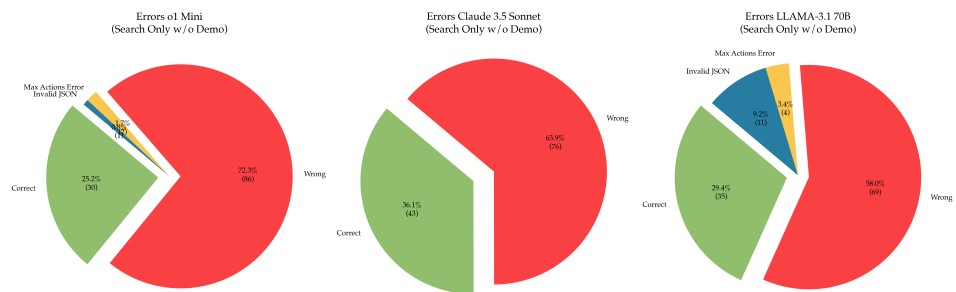

Figure 13: Different technical errors for the CiteAgent with `Search Only` command without Demo comparing the o1-Mini, Claude 3.5 Sonnet and LLaMA-3.1 70B backbone.

# G   Price and Duration Distribution

In this section, we break down runtimes and costs associated with running CiteAgent with a GPT-4o or Claude 3 Opus backbone.

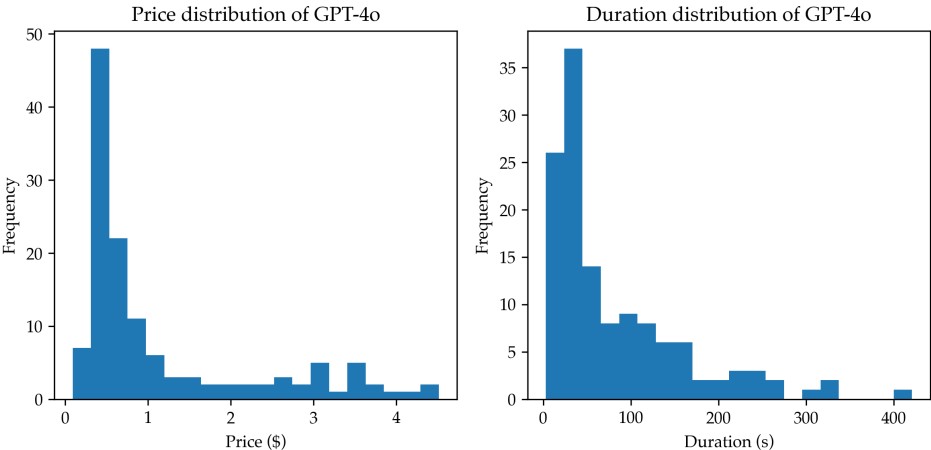

Figure 14: Price and duration distribution on CiteME with the `Read` and `Search` command with Demo for the GPT-4o backbone. The average price is ∼$1.2 per run or ∼$150 in total. The average duration is $82.9 \, \text{s}$ per citation or $10772 \, \text{s}$ in total.

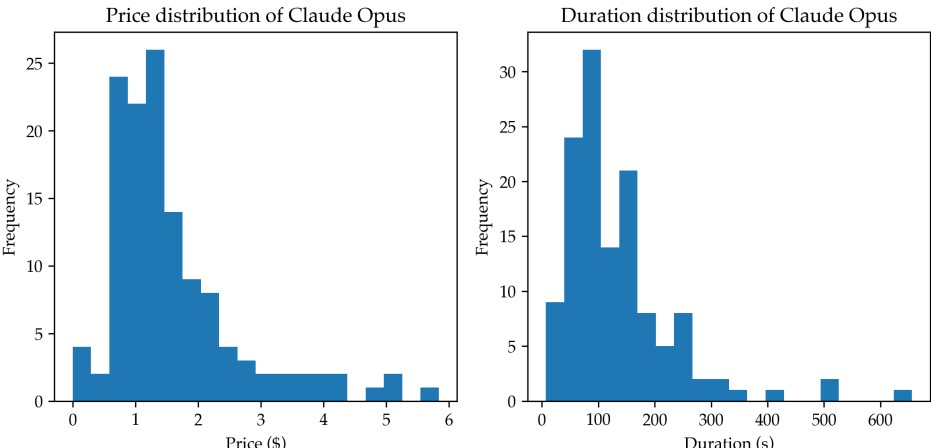

Figure 15: Price and duration distribution on CiteME with the `Read` and `Search` command with Demo for the Claude Opus backbone. The average price is ∼$1.6 per run or ∼$206 in total. The average duration is $136.0 \, \text{s}$ per citation or $17675 \, \text{s}$ in total.

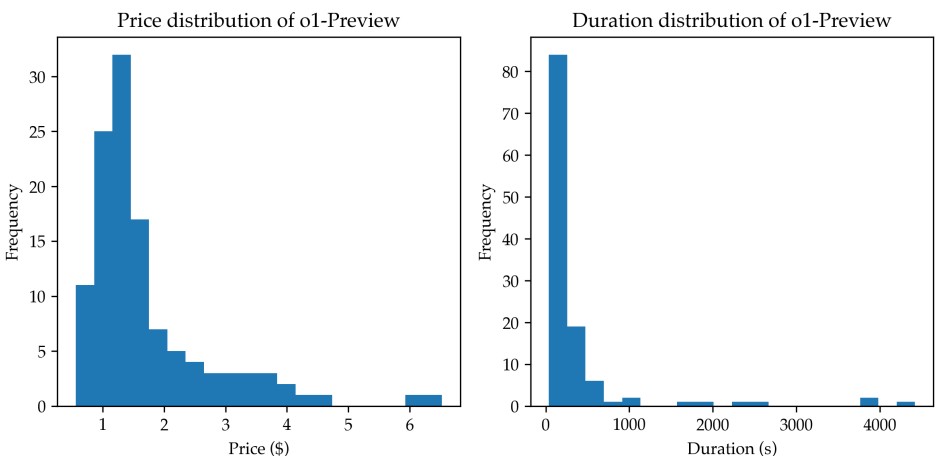

Figure 16: Price and duration distribution on CiteME with the `Read` and `Search` command with Demo for the o1-Preview backbone. The average price is ∼$1.7 per run or ∼$205 in total. The average duration is $369.8$ s per citation or $44006$ s in total.

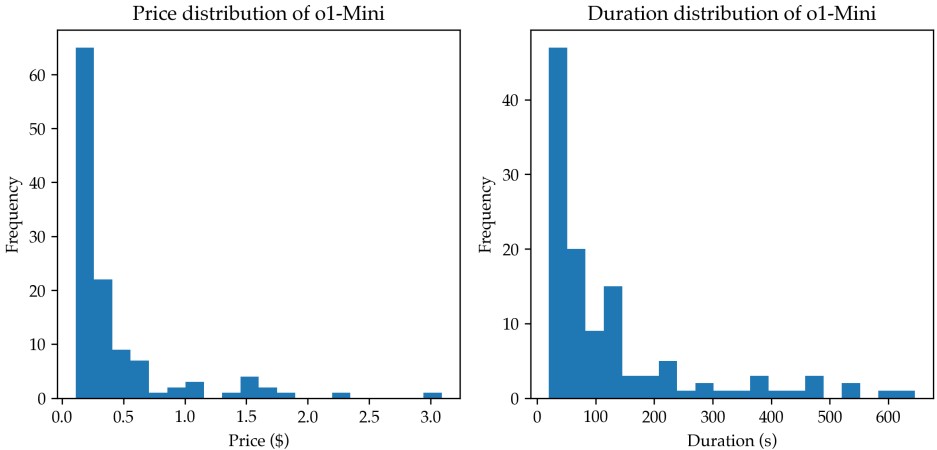

Figure 17: Price and duration distribution on CiteME with the `Read` and `Search` command with Demo for the 01-Mini backbone. The average price is ∼$0.4 per run or ∼$50 in total. The average duration is $125.1$ s per citation or $14886$ s in total.

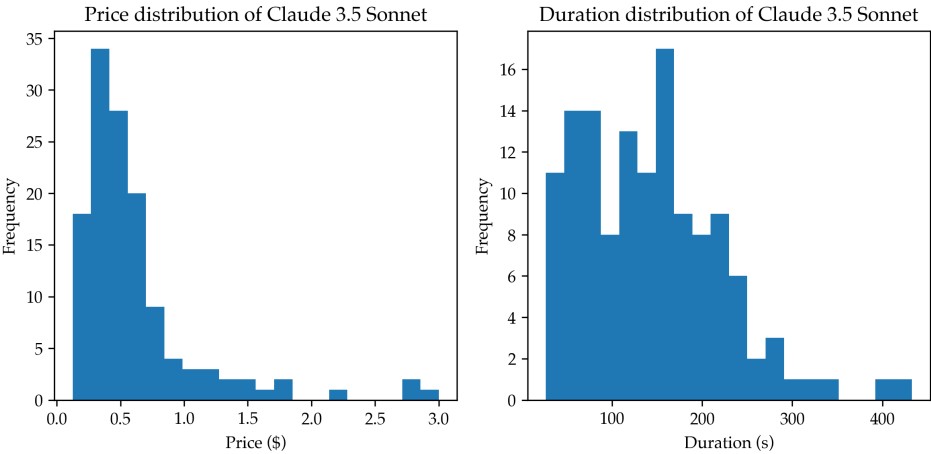

Figure 18: Price and duration distribution on CiteME with the `Read` and `Search` command with Demo for the Claude 3.5 Sonnet backbone. The average price is ∼$0.6 per run or ∼$80 in total. The average duration is 143.7 s per citation or 18686 s in total.

