# OpenReview forum: "CiteME: Can Language Models Accurately Cite Scientific Claims?"
_NeurIPS.cc/2024/Datasets_and_Benchmarks_Track — NeurIPS 2024 Track Datasets and Benchmarks Poster_

### Official Review · Reviewer_L4JP · 2024-07-08
**Benchmark on Citation Attribution**

**Rating:** 6
**Confidence:** 4
**Correctness:** Yes
**Clarity:** Yes

**Review:**

Originality:
* The proposed CiteMe benchmark evaluates a distinct, but potentially useful, capability of LLM tools and is intentionally constructed.
* The paper makes good distinction from related work in citation attribution datasets and context-aware recommendation. This could be improved by adding more detail as to how CiteMe and CiteAgent can improve scientific assistants.

Quality:
* The experiments evaluate modern LLMs with a standardized set of tools on the CiteMe tasks. The dimensions of the experiments are clearly defined and provide useful insights into the capabilities of LLMs.
* The error analysis provides additional insight into how to think of LLMs capabilities related to the CiteMe task.

Significance:
* The CiteAgent includes relevant tools for scientific assistants and could be useful for further applications.
* The design of the task could be better motivated. Given that many papers are referenced directly with citations, why is it important for an LLM to be able to perform this task in this way?

Clarity:
* The paper is generally well written and provides detailed description with relevant examples.

**Strengths:**

The paper has the following strengths:
* Clarity of writing and of curation for the benchmark with detailed description of the experiments.
* Potentially useful benchmark (CiteMe) and framework for enabling LLMs to performs citation attribution (CiteAgent).

**Additional Feedback:**

N/A

**Documentation:**

Yes

**Limitations:**

A brief statement of limitations was included in the appendix. I think it can be expanded to include more details of what the consequences of getting attribution right and wrong is and what the benchmark aims to measure and how that translates into relevant applications.

**Opportunities For Improvement:**

The paper could improve by addressing the following:
* Motivating the design of the CiteMe task, especially as it comes to making better scientific assistants. How does better performance on CiteMe lead to LLMs becoming better scientific assistants? It might help to look at papers that propose scientific assistant frameworks for scientific tasks and draw a connection to them [1] [2] [3].
* Why does Specter and Specter2 only have results for the "No-Commands" setting? Could you also clarify the reasons for choosing SPECTER models as baselines? It might be good to include an additional discussion on scientific language models.

[1] M. Bran, Andres, Sam Cox, Oliver Schilter, Carlo Baldassari, Andrew D. White, and Philippe Schwaller. "Augmenting large language models with chemistry tools." Nature Machine Intelligence (2024): 1-11.

[2] Boiko, Daniil A., Robert MacKnight, Ben Kline, and Gabe Gomes. "Autonomous chemical research with large language models." Nature 624, no. 7992 (2023): 570-578.

[3] Miret, Santiago, and N. M. Krishnan. "Are LLMs Ready for Real-World Materials Discovery?." arXiv preprint arXiv:2402.05200 (2024).

**Relation To Prior Work:**

Yes - this could be improved by adding more details related to applications of scientific assistants and adding relevant work. In addition to the examples mentioned above, here is some additional work. [1]

[1] Lála, J., O'Donoghue, O., Shtedritski, A., Cox, S., Rodriques, S.G. and White, A.D., 2023. Paperqa: Retrieval-augmented generative agent for scientific research. arXiv preprint arXiv:2312.07559.

**Summary And Contributions:**

The paper proposes a new benchmark on citation contributions, which includes 130 questions curated by human experts on citation contributions that are evaluated along four dimensions: Attributable vs Unattributable, Unambiguous vs Ambiguous, Non-Trivial vs Trivial, Reasonable vs Unreasonable. The dimensions aim to make the citation attribution as clear and tractable as possible. In addition to the benchmark, the paper also proposes a CiteAgent which includes modern LLMs interacting with tools (e.g., the Semantic Scholar API) to evaluate the performance on the CiteMe benchmark.

---

> ### Author Response · Authors · 2024-08-16
>
> We thank the reviewer for their detailed feedback. We greatly appreciate your recognition of the originality of our writing and the detailed curation of the benchmark, as well as the strength of our experiments, which you highlighted as being clearly defined and insightful.
>
>
> >Motivating the design of the CiteMe task, especially as it comes to making better scientific assistants. How does better performance on CiteMe lead to LLMs becoming better scientific assistants? It might help to look at papers that propose scientific assistant frameworks for scientific tasks and draw a connection to them [1] [2] [3].
> >This could be improved by adding more details related to applications of scientific assistants and adding relevant work. In addition to the examples mentioned above, here is some additional work. [1]
>
>
> Thank you for bringing these points. We have added the papers you mentioned to our discussion, as they are well connected to CiteME.
>
> [1] equips an LM with 18 tools, for the purposes of solving chemistry related tasks. One tool allows the LM to search for related work. CiteME offers the ability to benchmark this tool, in a standalone setting. An improvement by that tool on CiteME would potentially lead to an overall improvement of this entire system. This also applies to the systems presented in [2, 3]. Finally, we agree that PaperQA [4] is related to our work, but there’s a substantial difference in objectives. PaperQA poses biomedical questions which require reading a paper to answer, while in our dataset the task is to find a paper being cited. We’ve added this discussion to our related work section.
>
>
> Outside of academic settings, paper discovery tools such as undermind.ai, scholarai.io, researchrabbit.ai are already widely used today, and CiteME enables us to properly benchmark these systems. For example, we manually tested a few examples from CiteME and found undermind.ai’s system to not be able to correctly answer them [5]. Benchmarks like CiteME could be used to improve these systems, which real scientists use.
>
> Given the community’s excitement around literature exploration models, in both academic and commercial settings, we think that CiteME is highly relevant to improving the ability of LMs and other systems in literature recommendation.
>
>
> > Why does Specter and Specter2 only have results for the "No-Commands" setting? Could you also clarify the reasons for choosing SPECTER models as baselines? It might be good to include an additional discussion on scientific language models.
>
>
> Thank you for the great question! The two SPECTER systems are not language models but state-of-the-art retrieval systems for citation attribution. So the commands for LM-based agents are not applicable for SPECTER. We choose these models as a baseline because they are the best current approach in the literature for citation attribution. The very low score that these models obtain on our benchmark shows how challenging our task is.
> We will elaborate in the next draft on the differences between retrieval systems and our agent-based approach to citation attribution and discuss how scientific LMs could be used to enhance the agent-based approach.
>
>
>
>
>
>
> [1] M. Bran, Andres, Sam Cox, Oliver Schilter, Carlo Baldassari, Andrew D. White, and Philippe Schwaller. "Augmenting large language models with chemistry tools." Nature Machine Intelligence (2024): 1-11.
>
> [2] Boiko, Daniil A., Robert MacKnight, Ben Kline, and Gabe Gomes. "Autonomous chemical research with large language models." Nature 624, no. 7992 (2023): 570-578.
>
> [3] Miret, Santiago, and N. M. Krishnan. "Are LLMs Ready for Real-World Materials Discovery?." arXiv preprint arXiv:2402.05200 (2024).
>
> [4] Lála, J., O'Donoghue, O., Shtedritski, A., Cox, S., Rodriques, S.G. and White, A.D., 2023. Paperqa: Retrieval-augmented generative agent for scientific research. arXiv preprint arXiv:2312.07559.
>
>
> [5]
> - https://www.undermind.ai/query_app/display_one_search/34f8883ca84d1248c869a10ed5df0ee1c458ca4e63052e20ff39725e565a9263/
> - https://www.undermind.ai/query_app/display_one_search/5bfd5b25c1ad7ed41a8afc5a27ff2e0d91dd7c030063b17c972634c0bc55a0b4/
> - https://www.undermind.ai/query_app/display_one_search/2a85addf2621c86d8797e835793f64ff65d68711e8d636e4e6e91ded127df1f6/

---

> > ### Comment · Reviewer_L4JP · 2024-08-22
> >
> > Thank you for the additional details. They address most of my concerns and have adjusted my score. I still think that the CiteMe could be better motivated by pointing to direct applications. Paper discovery tools help motivate the benchmark, but it would be nice to see how that can lead to something else. For example, how does paper discovery help scientific discovery and how can models trained in CiteMe help facilitate?

---

> > ### Author Response · Authors · 2024-08-22
> >
> > Thanks for updating your score.
> >
> > [1,2,3] show that paper discovery is a crucial component of systems that automate scientific research. CiteME plays an important role in developing better tools for paper discovery, and provides a way to effectively measure their efficiency. Currently, these systems are tested as a whole, without isolating the tools responsible for scientific discovery. **CiteME allows us to evaluate components within them independently -- and we discover that current LM Agents are not yet ready for automated paper discovery, leading to serious gaps in end-to-end automated research pipelines.**
> >
> > In addition, most existing LM benchmarks are saturated, with most LMs scoring 80-95% on them. There is a need in the AI community to show what properties LMs currently lack, to show LM developers what aspects they should work on. On CiteME, the best LMs get less than 40%, clearly indicating to developers an important task that they could improve LMs on,  while also providing an indicator they can use to track progress.
> >
> >
> >
> >
> >
> > Please tell us if this addresses your concern. We will add these points to the next version of the draft.
> >
> >
> >
> >
> >
> > [1] M. Bran, Andres, Sam Cox, Oliver Schilter, Carlo Baldassari, Andrew D. White, and Philippe Schwaller. "Augmenting large language models with chemistry tools." Nature Machine Intelligence (2024): 1-11.
> >
> > [2] Boiko, Daniil A., Robert MacKnight, Ben Kline, and Gabe Gomes. "Autonomous chemical research with large language models." Nature 624, no. 7992 (2023): 570-578.
> >
> > [3] Miret, Santiago, and N. M. Krishnan. "Are LLMs Ready for Real-World Materials Discovery?." arXiv preprint arXiv:2402.05200 (2024).

---

> > > ### Comment · Reviewer_L4JP · 2024-08-27
> > >
> > > Thank you for the additional details. I think that most of my concerns are addressed.

---

### Official Review · Reviewer_L2LE · 2024-07-22
**Review for Submission2082**

**Rating:** 6
**Confidence:** 3
**Correctness:** Correct

**Review:**

LLMs are playing a more and more important role in academia, including paper writing, literature review, peer review etc. This paper addresses the issue of LLM assistance in literature review, specifically focusing on citation retrieval. This type of work is of importance. Refer to the "Strengths" and "Areas for Improvements" sections for detailed pros and cons.

**Strengths:**

- The paper introduces the first manually curated citation attribution benchmark, addressing the limitations of existing automatically curated benchmarks by eliminating unattributable, ambiguous, trivial, and unreasonable features. The benchmark's size is adequate for evaluating LMs.

- They propose an LM-based system, CiteAgent, to search the citations based on the benchmark. They include different commands, like search, read, and select in the system, and use various studies to validate the effectiveness of the commands.

- The paper analyzes the failed and successful cases of the agent. This helps to better understand how the agent works and learn the areas for improvement.

**Additional Feedback:**

NA

**Clarity:**

The paper is well-written and easy to follow. However, the appendix appears to have some formatting issues and is hard to follow.

**Documentation:**

The collection of datasets is reasonable and sufficient, and the provided dataset is also well-organized.

**Ethics:**

The paper has no ethical concerns.

**Limitations:**

The paper discusses the limitations and future work of the paper, like expanding the scope of the benchmark.

**Opportunities For Improvement:**

- In the human evaluation, the paper employs 20 experts. It would be beneficial to clarify whether these experts are machine learning domain specialists and if they are familiar with the source papers of the citation text excerpts. Additionally, it might be interesting to compare the LM agent's performance against humans with different levels of expertise.

- While the metrics used in the benchmarks appear reasonable, it would be helpful to explain why these specific metrics were chosen.

- Much of the evaluation relies on human judgment, such as the curation and labeling of text excerpts in Table 1. It would be beneficial to include inter-rater agreement metrics to assess the consistency of human evaluations.

- Given the input limits of the models, would it be possible to only input the abstract for the agent to read?

**Relation To Prior Work:**

The paper provides a clear and concise discussion of related work. However, since the benchmark relies on four popular citation prediction benchmarks, it would be helpful to include more details about these benchmarks, such as their size and domain.

**Summary And Contributions:**

This paper builds a benchmark, CiteME, that evaluates the abilities of LMs in citation attribution. Using this benchmark, the authors reveal a large gap between frontier LMs and human performance. To close this gap, they introduce CiteAgent, an autonomous system built on the GPT-4o LM that can also search and read papers, which achieves an accuracy of 35.3% on CiteME. They also analyze different setups of CiteAgent to show its effectiveness.

---

> ### Author Response · Authors · 2024-08-16
>
> We thank the reviewer for their detailed feedback.  We’re glad you found our benchmark CiteME, as well as our agent, CiteAgent to be strong points. We are happy to incorporate the suggested improvements by the reviewer, and detail our efforts below:
>
>
> > In the human evaluation, the paper employs 20 experts. It would be beneficial to clarify whether these experts are machine learning domain specialists and if they are familiar with the source papers of the citation text excerpts. Additionally, it might be interesting to compare the LM agent's performance against humans with different levels of expertise.
>
> All of our annotators are machine learning PhD students. Based on a survey we ran post-annotation, we estimate that the annotators were familiar with the paper being cited about a tenth of the time. This makes sense since our papers cover a wide variety of ML topics, while each annotator specializes in a single topic. We agree that comparing the agent’s performance against humans with different levels of expertise would be an interesting experiment to run, and we leave that for future work, as our main focus is on building a strong benchmark and baseline citation attribution systems.
>
> >While the metrics used in the benchmarks appear reasonable, it would be helpful to explain why these specific metrics were chosen.
>
>
> We are not sure which labels you are referring to, and would be grateful if you could clarify. If you are referring to the attributable vs. unattributable, ambiguous vs unambiguous, trivial vs non-trivial, and reasonable vs unreasonable labels: we first looked at previous citation attribution benchmarks  and tried to determine the properties that could make the excerpts in them of low quality. We noticed that these datasets have many excerpts that are either impossible to attribute or trivial to attribute. We believe that those two properties lead to bad benchmarks. And so we developed those labels shown above (unattributable, ambiguous and trivial) to more formally ascertain the quality of each benchmark. We have made this clearer in the revision.
>
> If you are referring to something else, we would be grateful to answer that as well.
>
>
> >Much of the evaluation relies on human judgment, such as the curation and labeling of text excerpts in Table 1. It would be beneficial to include inter-rater agreement metrics to assess the consistency of human evaluations.
>
> Thank you for this suggestion. We ran additional human experiments to measure inter-rater agreement. We gave five new human experts ten questions from CiteME, and found the following:
>
> 71% of the time, the new experts and the old experts agreed on the paper.
> Additionally, only 7% of the time, neither expert got the correct answer. We are now in the process of running this survey on the entire dataset and will add these results to the draft once the survey is complete.
>
>
> >The paper provides a clear and concise discussion of related work. However, since the benchmark relies on four popular citation prediction benchmarks, it would be helpful to include more details about these benchmarks, such as their size and domain.
>
>
> Thank you for the valuable feedback. We agree that providing more details about the other benchmarks we discuss in the paper will help clarify our work. We provide a comparison table below (and will highlight it in the next version of our draft):
>
> | Dataset | Query Set Size | Full Paper Text | Date Range | Domain|
> |-----------------------------|-----------------|-----------------|---------------| ---------------|
> | FullTextPeerRead [Jeong 2020]| 5K | ❌ | '07-'17 | Artificial Intelligence |
> | ACL-200 [Bird 2008] | 20K | ❌ | '09-'15 | NLP (ACL Venue) |
> | RefSeer [Huang 2014] | 625K | ❌ | Unk - '14 |  Engineering |
> | arXiv [Gu 2022] | 1.7M | ❌ | '91 - '20 | No domain restriction |
>  | CiteMe (Ours) | 218M | ✅ | '08 - ‘24 | Machine Learning (Including Computer Vision, NLP, Theory) |
>
>
> Note that our query set size is 100x larger than the largest previous dataset, which adds significant difficulty.
>
>
> This table emphasize the strengths of our benchmark compared to previous work. We will include this in the revised draft for added clarity.
>
> >Given the input limits of the models, would it be possible to only input the abstract for the agent to read?
>
> Thank you for the great question! We have already compared LM agents in the restricted "Search Only" setting, where only titles and abstracts are accessible. This is introduced in lines 195-198, with results shown in Table 3. For example, GPT-4o’s performance degrades by almost 6% when the ability to read papers is taken away from it. We will emphasize this point more clearly in the revised draft to enhance clarity.

---

> ### Author Response · Authors · 2024-08-28
>
> As the review period comes to an end, we would be grateful to know if our answers satisfied your questions, and if you have any other concerns.
> Thanks!

---

> ### Comment · Reviewer_L2LE · 2024-08-30
>
> Thanks for the author's detailed response. The answer to the metrics is what I am asking about, thanks for providing the detailed answer. It would be helpful to include it in the revised paper. Most of my questions are addressed, and I will increase my confidence.

---

### Official Review · Reviewer_m9VH · 2024-07-29
**a benchmark for correctly citing scientific articles**

**Rating:** 6
**Confidence:** 4
**Correctness:** The claims of authors seem to be gene…
**Clarity:** The structure and contribution of the…

**Review:**

The motivation is clear, and authors prove that the benchmark is challenging with several commercial LLMs and SOTa systems in use (SPECTER). Given that human performance is also under 70%, it seems that the proposed benchmark is suitable for investigating the LLM capability on citing proper articles, especially in ML area. However, considering the time variance of article citation capability depending on the LM used, it seems that the evaluation experiment requires more considerate settings.

**Strengths:**

Authors propose a challenging benchmark on citation attribution, which is proved to be quite challenging, evaluated with simultaneously proposed system CiteAgent with three popular commercial LLM backbones and SOTA system. Also, authors state rationales for setting four criteria in detail, to support the newly construction of the proposed dataset.

**Additional Feedback:**

(Updated score after author response)

**Documentation:**

The detail of experiment and dataset collection/cleansing phase need to be supplemented at least later in appendix.

**Ethics:**

No specific ethical concerns.

**Limitations:**

Some limitations I think are listed above: the quantity of excerpts and sampling method for inducing criteria for citation attribution dataset construction, and deeper discussion on the performance outdated or closed commercial APIs/LLMs.

**Opportunities For Improvement:**

Given that citation attribution is quite contextual task, authors seem to struggle in setting their criteria of the newly dataset construction. Some points I found opportunities for improvement are:
- Was 10 excerpts from existing datasets sufficient for discerning the points to improve in the literature? If this process was qualitatively done, I wonder if there was any sampling criteria for choosing the excerpts.
- Given that the number of articles that are published in ML area is soaring, the LM trained with latest information would be advantageous in overall evaluation. For instance, as far as I know, the appearance of Llama 3, Claude 3 Opus, and GPT-4o are subsequent, and some results seem to represent such difference. Also, given that SPECTER series seem to be updated more than a year ago, which adds limitation to be used as a backbone; which requires some discussion for the performance degradation.

**Relation To Prior Work:**

The paper adequately discusses the previous work.

**Summary And Contributions:**

This paper proposes CiteME, a benchmark that consists of excerpts from recent ML papers, each referencing single other paper, which results in LM accuracy of 18.5% and humans 69.7%, also suggesting CiteAgent an autonomous system based on GPT-4o that achieves 35.3% performance. Authors conduct a comprehensive experiment and error analysis using various commercial LLMs (though some results not available on popular open-sourced LLM llama series), finding out that SOTA systems fail to (even 0%) correctly cite desired articles.

---

> ### Author Response · Authors · 2024-08-16
>
> We thank the reviewer for their detailed feedback. We are glad that you find CiteME to be well-motived, challenging, and a suitable benchmark.
>
> > Was 10 excerpts from existing datasets sufficient for discerning the points to improve in the literature? If this process was qualitatively done, I wonder if there was any sampling criteria for choosing the excerpts.
>
> > the quantity of excerpts and sampling method for inducing criteria for citation attribution dataset construction
>
> We appreciate your suggestion that 10 samples (which we randomly selected) might not be sufficient to justify our claim about existing benchmarks,
> Therefore recompute the table with 50 random samples per dataset.  Our new results [below] on these larger samples are consistent with the previous results, thereby supporting our claim that across all four existing datasets, the vast majority (80%) of samples are ambiguous, trivial, or unattributable.
>
> | Dataset           | Reasonable [%] | Ambiguous [%] | Unattributable [%] | Trivial [%] |
> |-------------------|----------------|---------------|--------------------|-------------|
> | FullTextPeerRead  | 24             | 26            | 34                 | 16          |
> | ACL-200           | 26             | 42            | 18                 | 14          |
> | RefSeer           | 24             | 28            | 32                 | 16          |
> | arXiv             | 10             | 50            | 30                 | 10          |
> | **New Average**   | **21**         | **36.5**      | **28.5**           | **14**      |
> | **Old Average**   | **20**         | **37.5**      | **25**             | **17.5**    |
>
>
>
> To address the concern further, in addition to the manual analysis above, we conducted an automated analysis of the ambiguous category. Specifically, we identified excerpts that cited multiple papers simultaneously (e.g., \cite{paper1, paper2, paper3}) where one of the cited papers is the target. This analysis allowed us to establish a lower bound on ambiguous excerpts across all benchmarks (below). These excerpts can not serve well as questions since they have multiple different correct answers, whereas the respective benchmarks only include one correct target answer (as in CiteME).
>
>
> | Dataset | Ambiguous [%] |
> |---------|---------------|
> | arXiv   | 54.96         |
> | ACL     | 27.20         |
> | RefSeer | 12.61         |
>
>
> We note that this is just a lower bound estimate, as the automatic parsing is only able to detect a subset of the ambiguous excerpts. Still, these findings are consistent with our previous results, and show that previous benchmarks contain vast quantities of ambiguous excerpts.
>
> FulllTextPeerRead automatically deletes all other citations, so this was not possible to do in their case.
> We have updated Table 1 in the revised draft with the results with the expanded 50-sample sets and included the automatic evaluation data.
>
>
> >Weakness 2: Given that the number of articles that are published in ML area is soaring, the LM trained with the latest information would be advantageous in overall evaluation. For instance, as far as I know, the appearance of Llama 3, Claude 3 Opus, and GPT-4o are subsequent, and some results seem to represent such difference.
>
>
> > deeper discussion on the performance of outdated or closed commercial APIs/LLMs.
>
>
> The hypothesis you pose is interesting: Language models (LMs) may perform better on papers they encountered during training, with a drop in performance on newer papers, leading to better performance from more recently released models. To test this, we compare the results of our CiteAgent on excerpts from papers published before 2024 versus on excerpts from papers published in 2024. We note that the cutoff dates for Claude 3 Opus, Claude 3.5 Sonnet and GPT-4o are August 2023, August 2023 and October 2023 respectively.
>
>
> |                    | Before 2024 | 2024   |
> | ------------------------ | ----------- | ------ |
> | CiteAgent + GPT-4         | 36.99%      | 32.61% |
> | CiteAgent + Claude 3 Opus | 28.77%      | 21.74% |
> | CiteAgent + Claude 3.5 Sonnet | 42.47%      | 36.96% |
>
> This result shows that your hypothesis might be right, and that models perform better on papers which they observed during training.
>
> We will include this insightful analysis in our revised draft.

---

> > ### Author Response · Authors · 2024-08-16
> >
> > >Also, given that SPECTER series seems to be updated more than a year ago, which adds limitations to be used as a backbone; which requires some discussion for the performance degradation.
> >
> >
> > Thank you for raising this point. We believe there may be a small misunderstanding regarding the SPECTER models. SPECTER is not a language model but rather the state-of-the-art  retrieval system for citation attribution. It was trained only on a corpus of titles and abstracts and has never directly seen the text excerpts.
> >
> > Our SPECTER2 pipeline works by embedding all 200 million paper abstracts and titles in the Semantic Scholar database and performing an exhaustive similarity search to predict citations. This pipeline includes all the papers in CiteME, including the most recent papers from 2024.
> >
> > The fact that SPECTER2, a state-of-the-art retrieval system for citation prediction, performs poorly (with a score of 0) demonstrates the difficulty of our task for traditional retrieval systems. This challenge motivated us to explore agentic LM models as a more promising approach.
> >
> >
> > > Given that citation attribution is quite contextual task, authors seem to struggle in setting their criteria of the newly dataset construction.
> >
> > We agree- citation attribution is a task that requires a tremendous amount of context to perform correctly. We believe that our dataset formulation accounts for this and is a great way to benchmark models’ ability to understand context. We also believe that our criteria for constructing the dataset led to a good outcome, as human annotators have a 69.7%, while not spending more than 2 minutes per question. This is consistent with human annotators in a related setting [1].
> >
> > To further validate this, we randomly selected 50 samples from CiteME and repeated the human baseline, asking new human experts to predict the citation based on the excerpt. In 71% of cases, the new experts and previous experts agreed on the correct citation. Additionally, both experts failed to find the correct paper only 7% of the time. This suggests that the excerpts and their corresponding citations are relatively straightforward, making it a suitable ground truth for benchmarking LM Agent performance.
> >
> >
> > [1] Lála, J., O'Donoghue, O., Shtedritski, A., Cox, S., Rodriques, S.G. and White, A.D., 2023. Paperqa: Retrieval-augmented generative agent for scientific research. arXiv preprint arXiv:2312.07559.

---

> > > ### Comment · Reviewer_m9VH · 2024-08-25
> > > **thank you for rigorous response**
> > >
> > > Thanks authors for providing sincere response. I have read the response that resolves my original questions, and interestingly read newly provided results. Hope the rebuttal and response phase may improve the quality of the overall manuscript. I updated the score.

---

### Author Response · Authors · 2024-08-31

We thank the reviewers and want to summarize the rebuttal process:

Reviewers found our benchmark well-motivated (m9VH, L4JP), as well as novel and important (L2LE, L4JP). Here is a summary of the changes made during the rebuttal period, which will be included in the updated draft:

* **Larger Sampling of Previous Datasets**: Our larger analysis reconfirmed that *80%* of previous benchmarks contain ambiguous, unattributable, or trivial excerpts, critically impact their evaluation capability and motivating the necessity of CiteME.
* **Time-Sensitivity of Benchmarks**: Our analysis showed models perform up to *6% better* on papers cited before 2024 (that were part of their training data).
* **Comparison Table**: We added a table comparing CiteME with previous benchmarks, highlighting that CiteME offers *a 100x larger* query set and access to *full-text* of papers.
* **Context in Automated Research Pipelines**: We expanded on how CiteME evaluates automated paper discovery, revealing *critical gaps* in end-to-end automated research pipelines.

We appreciate the reviewers' confirmation that our response addressed their concerns, and that they have no further questions.
Thank you for the insightful feedback and productive discussion.

---

### Decision · Program_Chairs · 2024-09-26

**Decision:**

Accept (Poster)

**Comment:**

Summary:

This paper evaluates LM performance in a task the authors call 'citation attribution' - given a bit of text referencing a paper, finding the title of that paper. The authors introduce a manually curated dataset for evaluating this task, called CiteMe, and an agent called CiteAgent that carries out the citation attribution task using the Internet.

Contributions:

a. a new benchmark for citation attribution, CiteMe, that was humanly created, from which easy cases were removed, and human evaluated to make sure the citations are solvable.
b. a new 'citation agent' able to carry out the search using internet interrogation. The citation agent achieves moderate success, particularly in the version using GPT-4o

Summary of reviewers' opinions:

All reviewers felt that the paper was marginally above the threshold of acceptance. They felt CiteMe could be a useful resource, but questioned aspects of the proposal. E.g., m9VH questioned whether the number of examples used to assess LLM's ability at the task was sufficient (the authors replied by running a new evaluation with 50 examples instead of 10). Reviewer L2LE also questioned aspects of the method used to assess LLM's competence at citation attribution - e.g., the absence of an intercoder agreement test.

Summary of rebuttal:

As already mentioned, the authors responded to  m9VH by running a new assessment with 50 examples instead of 10. They did the same for a number of other questions - e.g., by computing percentage agreement on the human assessment of system performance.

Summary of strengths:

a. The CiteMe dataset would appear to be a useful resource.
b. The new citation agent could also be useful.
c. The work is described clearly (although see below).

Summary of weaknesses:

a. The description of the procedure used to construct the dataset is very sketchy.
b. A number of methodological issues were highlighted by the reviewers, but many were fixed since the original reviews.

Summary opinion:

This strikes me as a substantial piece of work. Questionable aspects of the methodology used were corrected after submission. Overall, the paper is a marginal accept but could be accepted.